# Improvement of Quality of Higher Education Institutions as a Basis for Improvement of Quality of Life

Zorica Lazić [1], Aleksandar Đorđević [1,*] and Albina Gazizulina [2]

1   Faculty of Engineering, University in Kragujevac, 34000 Kragujevac, Serbia; zoricalazic29@gmail.com
2   Institute of Advanced Manufacturing Technologies, Peter the Great St. Petersburg Polytechnic University, 195251 St. Petersburg, Russia; albinagazizulina@gmail.com
*   Correspondence: adjordjevic@kg.ac.rs

**Abstract:** This paper aims to propose a quality assessment model for higher education institutions in the technical–technological field and a system for decision support and optimal management strategies for quality improvement. Obtaining research results is based on surveying stakeholders in higher education and obtaining quantitative data regarding key performance indices. Quantitative data and the genetic algorithm method are applied to determine optimal management strategies for quality improvement. Quality in the higher education sector is among the current issues in the academic community. By monitoring and researching the higher education field and analysing the literature and the current situation in the system of higher education in developing countries, it can be concluded that there is no single way to assess the quality of higher education institutions. This knowledge was a good starting point for the research presented in this paper. Accordingly, the findings include developing a system for quality assessment and the ranking of higher education institutions. Additionally, evaluating the relevance of key performance indicators of higher education institutions differs from different stakeholder perspectives. However, it is possible to develop a system for decision support and the selection of the optimal strategy for improving the performance of study programs and higher education institutions with regard to quality. The practical implications include defining a decision support system that enables the adoption of optimal decisions by the management teams of higher education institutions to improve study programs and the performance of the higher education institutions. The presented system may enable the benchmarking, simulation, and verification of different scenarios for improving the quality and performance of higher education institutions. In this paper, the authors analysed the characteristics, benefits, and drawbacks of different ranking systems to develop and introduce a novel ranking system that suggests weights for the ranking criteria and different perspectives regarding new digital age requirements. The model was tested, and the results are presented to demonstrate the advantages of the developed model. The originality of the research lies in the presented novel model that can be made available to government institutions and serve as a basis for the overall ranking and evaluation of higher education institutions, with the possibility of developing a performance-based funding system. Additionally, other stakeholders can gain an insight into the performance of an institution in relation to their needs and goals.

**Keywords:** higher education institutions; ranking; value-based management; performance assessment; performance indicators

## 1. Introduction

The issue of the quality of higher education (HE) is among the most critical issues. This idea has been of great importance in recent years in the whole of Europe. Among the essential issues is the performance-based evaluation of HE and its use for different purposes, including public funding. Several EU countries have introduced performance-based systems, measurements, and the funding of higher education institutions (HEIs)



based on performances [1,2]. International practice shows numerous examples of HEI rankings [3–5]. However, within these systems, the existence of different stakeholders and different needs and expectations with regard to developing, measuring, and monitoring performance systems to assess the quality of HEIs have not been entirely considered [6,7]. For instance, the authors of [8] state that the Shanghai ranking list must be considered highly controversial and questionable. It cannot assess the complex issue of HEI quality and states that HEIs undertake changes that are usually highly controversial and subject to criticism from different stakeholders. Another vital ranking system is Forbes Magazine's America's Best Colleges, and this system introduces a new dimension, according to which student success is ranked during and after graduation. This ranking methodology's essence is to emphasise the strength of academic institutions through their former students' success in the labour market [9,10] and not through scientists' achievements within the institutions. Therefore, these examples have some shortcomings, which are primarily reflected in the one-dimensional experience of quality and its ranking.

In this context, quality assessment and monitoring through indicators are necessary management tools. The performance of each of the processes carried out at HEIs can be measured through key performance indicators (KPIs) [11]. For the purpose of this paper, the authors present a model for the quality assessment and ranking of Serbian HEIs and a decision support system to improve HEI performance in the technical and technological fields. Such a system could provide benefits for various stakeholders, including the possibility of performance-based financing and the satisfaction of the end-user in educational services [12]. The considered ranking methodologies represent an essential starting point for further research in the scientific world to improve HEI ranking systems.

According to [13–16], developing states encourage HEIs to achieve better results, specifically when it comes to strengthening their management capacity, information systems, and quality monitoring tools. Academic communities are innovating different ranking HEI methodologies, divided into two groups, based on academic and nonacademic criteria. The academic criteria imply the establishment of ranking criteria according to the achievements of HEIs themselves, i.e., the achievements for which the teachers and researchers are responsible [17,18]. In contrast, the nonacademic criteria are focused on current and former university students' achievements, i.e., alumni members' success.

A comprehensive search and analysis of the numerous literature were performed to define quality and an appropriate methodology [19]. Given that the HE concept of quality is not one-dimensional, there is no single definition that would encompass all of its connotations. The authors of [20] define quality using the following characteristics: quality as excellence, quality as compliance with standards, quality as a convenience, quality as value for money, and quality as transformation. Each group of education beneficiaries and stakeholders, i.e., students, parents, academia, employers, and the state, has a different perspective of quality. For example, students associate quality with their institution, chosen study program, modules, and diploma title after graduation. Students aim to ensure an advantage in the labour market concerning the competition with their education degree and achieve quality as excellence [21]. Employers deal with quality in terms of finished products, which are students with acquired competencies for the labour market from the education system. They aim to hire graduates who possess a high level of knowledge and skills to cope with business challenges and business complexity, seeking to gain an advantage over the competition [22]. In developing states, which are the main financiers of HE, there is a need for the resources to be used efficiently to achieve a satisfactory level of quality, that is, to achieve quality for the money invested [23]. In this way, developing countries may achieve part of their strategic goals and sustainable development with the help of education systems because as an individual progresses through quality education, the countries acquire a quality educated society [24]. The different views of the stakeholders on the quality of HE are sufficient to define new tasks, which is to define the criteria based on which the quality of HE is assessed. Therefore, to define quality in HEIs, all stakeholders should assess needs, requirements, and expectations of quality from different perspectives.

According to all these stakeholders' needs, requirements, and expectations, the authors of this paper attempt to define indicators and develop a system for measuring and evaluating the quality of HEIs [25].

Today, there are at least 50 national ranking systems and about ten sizeable international ranking systems. Generally, they have different goals such as: improving performance in core activities—higher quality (weeding out underperformers and increasing efficiency); enhancing accountability and transparency (informing stakeholders; supporting dialogue and trust); encouraging HEIs to position themselves strategically: diversify and align national and institutional policies and activities. In these contexts, the study was performed within the HE system of the Republic of Serbia. Consequently, the Republic of Serbia needs to interconnect its efforts with the EU environment and provide and use the best experience from the EU in the development of its model that will be compatible with the EU. On the other hand, the Serbian model could be used in other similar countries or as the basis for further EU model development or a joint EU model development. Serbia introduced the accreditation process by signing the Bologna Declaration in 2003, which obliged the signatory states to approach the quality assurance of their educational system responsibly, following the general principles and guidelines [11,26,27] of the Bologna process. In this way, Serbia took the first significant step to develop its HEIs ranking system [27]. There is a need for an approach that will include attitudes about quality, the academic community and students and employers, and government institutions.

By analysing the literature and the current situation in the HE system in developing countries, it can be concluded that there is an idea of the importance of study programs, HEIs, and education systems quality at all levels, even in the conditions of the COVID-19 pandemic [28]. The quality assessment of HEIs is a vital issue since it can be used for ranking educational institutions, allocating performance-based funding by the state, defining development strategies for improving HEIs, and completing social and economic development [29,30]. That is why the quality of HE and HEIs is at the centre of interest in international legislation [31].

It is evident that there are some approaches for ranking and evaluating HEIs and study programs—all of them have both advantages and disadvantages. In this paper, the authors present an overview of the most common ranking and evaluation systems to answer the following research questions:

- How objective is the ranking conducted by the most crucial ranking lists of universities globally?
- Is accreditation sufficient to assess the quality of HEIs in the Republic of Serbia?
- What are the different methodologies and criteria for assessing the quality of HE?

The advantages and disadvantages are explored as a basis for developing a new model for performance-based evaluation and the demands of specific stakeholders (government, employers, students, and the academic community). In the first step, KPIs are defined and grouped into specific "dimensions". Based on this indicator, the mathematical model for evaluation is presented, and finally, in the context of digitalisation, the software solution developed. In the last step, the developed system is tested with accurate data from HEIs in Serbia.

The study aims to present a model for assessing the quality of HEIs in the technical–technological field from aspects of various stakeholders. Furthermore, the study aims to present a system for supporting decision making and making optimal decisions on improving individual quality indicators to define appropriate management strategies and improve quality.

HE drives the development of any society since it drives the growth of the economy and represents the mainstay of achieving a successful career for each individual. Therefore, the main topic of this paper is the quality of HE, which is seen as an aspiration towards continuous improvement of all HEIs processes and their outcomes in achieving the ideal economy and society based on knowledge [32].

The authors of the paper paid attention to selecting the relevant variables and the empirical approaches that can be used to measure HEIs quality. More specifically, the authors bring innovation into the current academic literature in the field by combining different approaches, such as quantitative statistical methods and multiple-criteria optimisation, so as to define and develop the HEIs ranking model. The entire developed model can be put at the disposal of government institutions and serve as a basis to rank and evaluate HEIs in order to develop a performance-based funding system.

The expected results presented in the paper are both theoretical and practical. The expected theoretical results are reflected in the defined model for quality assessment and performance assessment (based on KPIs) of study programs and HEIs while considering a complex group of stakeholder requirements. This model opens up the way to define the methodology for measuring, monitoring, and improving the HEIs and the quality of the education system. The application result includes the definition of a decision support system that will enable optimal decision-making by management teams of the HEIs to improve performance at the level of study programs and HEIs. In addition to all of the above, it will be possible to conduct benchmarking, i.e., compare performance with the best in the class and learn from successful institutions. The developed system can enable the simulation and verification of different scenarios for improving the quality and performance of the system, which can yield highly positive effects on the management and improvement of the given HEIs.

As for the organisation of the paper, the authors introduce quality in HEIs from different stakeholder perspectives, performance evaluation and assessment issues in Section 2, and international and national ranking systems in Section 3. The authors describe KPIs divided into six dimensions in the Section 4. The metrics, mathematical model, and model application results are introduced in the Section 5. Section 6 introduces scientometrics as a platform that may be integrated with the presented mathematical model to obtain more comprehensive details on university researchers. Finally, Section 7 deals with the results and Section 8 finishes with concluding remarks.

## 2. Literature Review

Some questions could be raised in this field:

- To what extent is the objective ranking of universities conducted by the most influential organisation?
- Is the process of accreditation sufficient for the evaluation of the quality of HEIs?
- Is there a specific methodology that could be used for the evaluation of the quality of the HEIs?

Academic institutions have recently been hit by significant reforms aimed at improving their performance levels. These reforms are inspired by various factors, such as budget constraints imposed by national governments, HEIs comparison globally, and the marketing sector [33].

Through literature analysis in the field of HEIs quality assurance, shortcomings have been noticed which have not been treated sufficiently. These shortcomings are reflected in the lack of a unified quality assessment and ranking system for HEIs in the European Union countries [34]. One of the reasons for the lack of a unique quality assessment system may be found in the fact that the quality of a HEI has different meanings for different groups of education users and stakeholders. This fact is one reason why standard quality assessment models, such as service quality model (SERVQUAL) or other proposed models that are more or less one-dimensional, cannot be applied [35]. The authors of [36] state that HEIs are undergoing essential changes that involve developing new roles and missions, with implications for their structure. They also claim that there is difficulty establishing classification criteria for existing indicators for which there is no consensus.

Therefore, the need to develop a model for the HEIs quality assessment in the techno-technological field from different stakeholder perspectives has gained attention [37]. Although many quality assessment models are developed for the manufacturing and in-

dustrial sectors, they cannot be applied directly to the HE sector. Some previous studies showed that the nature of manufacturing and HE sectors are different [38,39]. Different models that include performance and their KPIs should be developed and tested to meet the requirements of HEIs [40,41].

The authors of [40] define quality as a multidimensional concept depending on different stakeholder views resulting in different quality and performance indices. Due to the vague concept of quality and different meanings for stakeholders and the complicated nature of the educational process and service [38], many authors find it challenging to manage the quality of HE. The information on the quality of study programs or HEIs and their status compared to other faculties and programs [42] becomes vital for choosing between many HEIs and study programs [43]. E-learning is becoming standard in the COVID-19 crisis, so it is essential to understand the impact of e-learning on society and its benefits. The authors of [44] aimed at finding the determinants of the user-perceived satisfaction, use, and personal impact of e-learning. The study proposed a theoretical model integrating theories dealing with satisfaction with information systems and success in the e-learning systems. The model was empirically validated in HEIs through a quantitative structural equation modelling method. The drivers of user-perceived satisfaction are information quality, system quality, instructor attitude toward e-learning, diversity in assessment, and learner perceived interaction with others. In the study [45], the authors attempted to mitigate the literature gap concerning e-learning and social media use for active collaborative learning and engagement and its effect on the learning performance of research students in Malaysia. The study concludes that overall, active collaborative learning and engagement through e-learning and social media enriches students' learning activities and facilitates group discussions, and hence, its use should be encouraged in the learning and teaching processes in HEIs.

Furthermore, in an educational context characterised by globalisation, reputation is a crucial issue for modern HEIs. The investigation results on the relationship between internationalisation and reputation in top HEIs reveal that internationalisation positively influences the reputation of a university and moderates the relationship between its reputation and its institutional performance concerning research quality, teaching quality and graduate employability [46].

Consequently, the need to define and evaluate performance indicators in HEIs arose primarily due to the need for measurable and objective quality indicators, which depended only on experts' opinion in the field until recently. It is necessary to include multiple sources of information to provide objective explanations that reflect all HE complexity in the HEIs quality assessment model [47]. The authors of [48] examined how Total Quality Management (TQM) principles and core concepts can be measured to provide a means of assessing the quality of different institutions on various aspects of their internal processes. It is found that the quality assurers could use the measurement method in the UK to assess the education quality of HEIs.

Special attention is drawn to applying assessment systems based on KPIs to monitor and measure the quality HEIs. Different types of indicators are distinguished in the literature. The development of a model for quality of HEIs assessment through a KPIs system creates a basis for comparison in developing countries and around the world [49]. The development and implementation of a KPIs assessment system could have a broader goal [50]—to improve the management, operation, and quality of HEIs and education systems globally. However, having reviewed the literature, it can be concluded that there is no unique system based on quality KPIs. According to the purpose and area of application in the literature, indicators are defined in different ways. It is clear that there is no universal metric or universal set of KPIs that would enable the objective evaluation of the quality and position of HEIs, but it is clear that there are numerous approaches that, to a greater or lesser extent, try to rank HEIs [51].

Most authors agree [52–54] that KPIs can be defined as measures that provide the context of information and statistics, enabling comparisons between different areas alongside

other accepted standards. KPIs provide information on how public stakeholders are satisfied with the institution and the entire HE sector and how education goals have been met within the institutions and the entire HE sector. KPIs can facilitate sustainable educational policy implementation, inform HEIs about possible problems, and determine some of the causes of the problems [55]. KPIs are becoming increasingly important since they enable monitoring and assessing the situation in various areas to conclude quality objectively [56]. Well-selected and measurable KPIs should be carefully defined and used to determine which parts must take specific actions, i.e., where quality can be improved [57]. Quality KPIs represent empirical information that gives a picture of how a HEI realises its goals and ensures continuous monitoring of its quality level [58]. KPIs should provide information on whether HEIs have achieved the planned results and achieved the set goals and the extent to which they deviate from the planned values. Quality indicators make it possible to monitor performance for comparison, facilitate institutional functioning assessment, and provide information for external quality evaluation needs [47].

Existing ranking systems of universities and HEIs are widely accepted. Students use these tools to select appropriate institutions [43] for their education, according to their interests and needs. Researchers use these tools to select institutions according to excellence in their research field, while universities and HEIs apply these tools to position themselves as the leaders in this service industry. Agasisti and Johnes [59] state that the development of rankings as a policy and managerial tool is of particular relevance. Global ranking systems trigger different public opinions, from positive (ranking summarises) [60] to critical ones [61] to highly negative opinions. With the emerging importance of ranking universities in HE systems, there has been a lot of misuse and manipulative actions appearing to improve the rankings of specific institutions. Some institutions misrepresented the number of teaching staff, students and even merged institutions to obtain better positions in specific university ranking systems [61]. Daraio, Bonaccorsi and Simar [62] list four main types of criticisms of university rankings, namely: monodimensionality, statistical robustness, dependence on university size and subject mix, and lack of consideration of the input–output structure.

The fundamental issue of global ranking systems is their influence on institutional strategy and policy. According to Hazelkorn [63], rankings are wrong since they usually present statistical information about education quality, with factorisation of elite universities with many students. These universities create a distorted image and sense of quality since they mainly focus on research neglecting the number of issues in education, teaching and learning [64]. Consequently, ranking systems should have an essential role in promoting more meaningful public information and disclosure, comparing performance internationally to inform students/parents, governments, and the broader public [64].

Ranking universities has become a serious business [65]. Global ranking systems started a discussion in the academic community and initiated investment in HE to improve the competitiveness of national educational systems. Baty [66] pointed out that no ranking system could cover all aspects since none of the ranking systems is objective, and consequently, the rankings are subjective, leading to the fact that the indicators are essential so that data could be compared. Some other scholars believe that methods and tools for analysing key quality indicators and benchmarking will be increasingly crucial for different stakeholders and for the ranking of even whole educational systems.

Devising indicators that encompass all HEIs and their various missions, clustering the results for developing countries, and refraining from comparing them with the results from high-income countries would be positive steps to enhance the ranking systems [67].

It is a clear fact that there are no reliable, public, international and comparable data about educational systems. Consequently, the authors and facilitators of different ranking systems are forced to measure and compare results, but not a full scale of results. They base their ranking systems mostly on traditional research input from bibliometric and citation databases such as Clarivate, Thompson ISI, Elsevier-Scopus.

Based on the literature review, the following hypotheses have been formulated:

**Hypothesis H1.** *It is possible to develop a system for quality assessment and ranking of higher education institutions.*

**Hypothesis H2.** *It is possible to develop a system to support decision-making and selection of the optimal strategy for improving the performance of study programs and higher education institutions from the aspect of quality.*

Based on the presented hypotheses, the following research model has been developed (Figure 1).

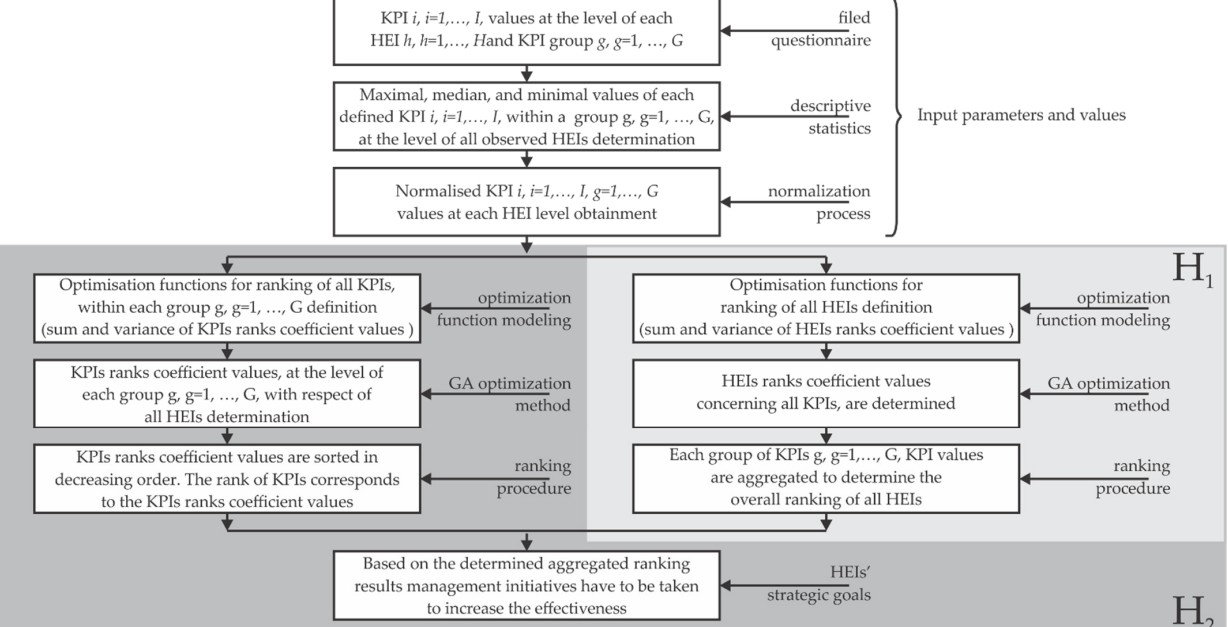

**Figure 1.** Research model

The methods and materials that serve as a starting point are further discussed to define specific research model parameters.

## 3. Materials and Methods

Global university ranking started in 2003, with the first year of Shanghai Ranking. Different ranking lists emerged and became increasingly popular among students and the academic community for a short period. Right now, different raking lists can be found, such as Shanghai Ranking (ARWU), Times HE, U-Multiranking, QS (Quacquarelli Symonds), Leiden Ranking, and иWebometrics (CSIC). In this paper, the most influential ones will be analysed in order to determine their advantages and disadvantages.

The Shanghai Ranking is one of the ranking systems for universities. The leading indicators are the number of Nobel Prize winners, leading scientists, and published articles in the leading journals. However, the repositories are mainly in English, so an institution with the main focus on social sciences and humanities is less visible than in other languages. Some authors [68] pointed that comparing a large university with a large budget, such as Harvard, with a small national university is the same as comparing a race car with small family cars. Another ranking system, Times HE (THE), is also based on the ranking by parameters from the educational process, research, knowledge transfer and international cooperation. The significant criticisms are that these rankings are focused only on some university activities. For instance, the Shanghai Ranking focuses on research, while THEfocuses on reputation and internationalisation [69]. Databases with questionnaires for students and employees used for ranking usually have a low level of input from

stakeholders [70]. Different institutions have different missions and goals [71], which is usually neglected in the ranking process. Institutions with different goals cannot be easily compared and ranked on the same list. It is essential to secure "clean" rankings, which are transparent, free of self-interest, and methodologically coherent, creating incentives for broad-based improvement [72].

Some more recent methodologies, such as U-Multiranking, are intended to improve the existing ranking system, introducing a multidimensional approach to compare and benchmark higher eduction institutions. This ranking system comprises complete institutions, study programs, and specific universities, such as teaching and learning, research, regional and international cooperation. National ranking systems tend to include a more comprehensive set of indicators [73].

Table 1 presents an overview of some international and national ranking systems and academic research, and we notice overlapping of the sets of indicators in the different systems. If Tables 2–4 are considered, it is possible to notice that most of the ranking systems have the same weight of indicators. For instance, ARWU and U-Multiranking cover the quality of education, institution, research, and productivity. U-Multiranking also considers teaching and learning, research, knowledge transfer, internationalisation and regional engagement indices. The main difference is the one-dimensionality of ARWU compared to the multidimensional approach of U-Multiranking. ARWU possesses indicators for only 500 universities, while U-Multiranking possesses many more (only 28% of institutions have a grade for the quality indicator in teaching in one field and 13% in three fields). Some national ranking systems, such as those used in Poland (PERSPEKTYWY) and North Macedonia (MACEDONIA HEIs), are based on two mentioned systems. North Macedonian HEIs use the ARWU system and PERSPEKTYWY is based on the USA ranking system, i.e., on the Best College Ranking. The ranking system in Saudi Arabia (HEQAM) is based on the model of SERVQUAL, using AHP for the identification of the priorities and weights of criteria and their alternatives. The model of evaluation is based on eight dimensions using known indicators. Different researchers such as Luzanin [74] introduce different dimensions and sets of indicators (53 indicators in the groups teaching and learning, quality of an institution, quality of education, research, knowledge transfer and international orientation). Tasic [75] selects indicators in model development for ranking HEIs developing a conceptual model to overcome the drawbacks of the existing models (model of 62 indicators in the following dimensions: education, research, cooperation with the economy, internationalisation and regional engagement). Petrusic [76] presents a model developed for Croatia with the following groups of indicators: quality of teaching, research, impact on society, and institutional quality mechanisms.

**Table 1.** Overview of used indicators in different ranking systems.

| Ranking Systems | Indicators | | | | | | | |
|---|---|---|---|---|---|---|---|---|
| | Quality of Education | Teaching and Learning | Quality of Faculties/Institutional Mechanisms in Institutions | Research | Knowledge Transfer | International Orientation/ Internationalization | Regional Engagement | Productivity |
| ARWU | x | | x | x | | | | x |
| U-MULTIRANK | | x | | x | x | x | x | |
| PERSPEKTYWY | | x | x | x | | x | | x |
| MACEDONIA HEIS | | x | | x | x | | | |
| HEQAM | | x | x | | | x | | |
| LUZANIN [74] | x | x | x | x | x | x | | |
| TASIC [75] | x | | x | x | x | x | x | |
| PETRUSIC [76] | | x | x | x | | x | x | |

**Table 2.** Comparison of used indicators of international ranking systems.

| Indicators | Ranking Systems | |
|---|---|---|
| | **ARWU** | **U-MULTIRANK** |
| Quality of education | Alumni of an institution winning Nobel Prizes and Fields Medals 10% | |
| Teaching and learning | | -Percentage of graduates in necessary studies<br>-Percentage of graduates in master studies<br>-Graduate on time in elementary studies<br>-Graduate on time in master studies |
| Quality of faculties/Institutional mechanisms in institutions | The staff of an institution winning Nobel Prizes and Fields Medals 20%<br>-Highly cited researchers in 21 broad subject categories 20% | |
| Research | -Papers published in Nature and Science* 20%<br>-Papers indexed in Science Citation Index-expanded and Social Science Citation Index 20% | -Number of quotes<br>-Total number of research publications<br>-Relative number of research publications<br>-Revenue from external research<br>-Artistic performance<br>-Number of citations in top publications<br>-Interdisciplinary publications<br>-Postdoctoral posts<br>-Professional publications<br>-Research partnership strategy<br>-School graduates |
| Knowledge transfer | | -Publications published in collaboration with industry<br>-Income from private sources<br>-Total number of patents published<br>-Relative number of published patents<br>-Published patents in collaboration with industry<br>-Number of spin-off companies<br>-Number of patent publications<br>-Revenue from continuing vocational training |
| International orientation/Internationalization | | -Frequency of foreign language study programs<br>-Frequency of foreign language master study programs<br>-Mobility of students<br>-Foreign citizens in professor status<br>-Joint international publications<br>-Number of international doctorates |
| Regional engagement | | -Number of undergraduate students enrolled in the region<br>-Number of graduates of master studies employed in the region<br>-Number of students on an internship in the region<br>-Number of joint publications in the region<br>-Revenue from regional sources<br>-Research partnership strategy in the region |
| Productivity | -Per capita academic performance of an institution 10% | |

**Table 3.** Comparison of used indicators of national ranking systems.

| Indicators | Ranking Systems | |
|---|---|---|
| | **PERSPEKTYWY** | **MACEDONIA HEIS** |
| Quality of education | -Prestige<br>-Academic reputation<br>-International recognition | |
| Teaching and learning | -Teaching staff<br>-Accreditation | -Percentage of students who have passed the state matriculation examination 5%<br>-Average number of credit students who have passed the state high school diploma 5%<br>-Percentage of international students 5%<br>-Academic staff/student ratio 4%<br>-Percentage of academic staff with the highest grade 8%<br>-Percentage of academic staff with one year or more work experience abroad 6%<br>-Percentage of students with scholarships from the Ministry of Education and Science 6%<br>-Institutional income per student 2%<br>-Library charges per student 1%<br>-Cost of IT infrastructure and equipment per student 1%<br>-Percentage of students who have graduated in full time 1%<br>-Percentage of undergraduate students with three months or more spent abroad due to foreign study/practical experience following national agreements 2%<br>-Employment rate after graduation 4% |
| Quality of faculties/Institutional mechanisms in institutions | -Study area—first and second degree<br>-Parameter estimation<br>-Academic staff with the highest qualifications<br>-Right to obtain a PhD<br>-Right to award a doctoral degree with facilitation<br>-Number of master study programs | |
| Research | -External research funding<br>-Development of faculty/teaching staff<br>-Awarded academic titles<br>-Publications<br>-Quotation<br>-FWCI (Field-Weighted Citation Impact) Index | -Total research revenue per academic staff 4%<br>-Revenue from research by the Ministry of Education and Science by academic staff 6%<br>-Papers published in peer-reviewed journals by academic staff 6%<br>-Papers indexed by Web of Science by academic staff 10%<br>-Books published by academic staff 4%<br>-Number of doctorates approved per academic staff 6% |
| Knowledge transfer | -Reputation of employers<br>-The economic situation of alumni | -Revenue from industry research by academic staff 6%<br>-Patents by academic staff |
| International orientation/Internationalization | -Programs in foreign languages<br>-Number of students studying in a foreign language<br>-Number of foreign students<br>-Foreign teachers<br>-Exchange students (number of students leaving)<br>-Exchange students (number of students to come)<br>-Multicultural structure of the total number of students | |
| Regional engagement | | |
| Productivity | | |

**Table 4.** Comparison of used indicators of different research works.

| Ranking Systems | Indicators | | | |
| --- | --- | --- | --- | --- |
| | **Quality of Education** | **Teaching and Learning** | **Quality of Faculties/Institutional Mechanisms in Institutions** | **Research** |
| HEQAM | | -Curricula<br>-Teaching staff<br>-The library | -Administrative services<br>-E-service<br>-Location<br>-Infrastructure | |
| Luzanin [74] | -International awards and scholarships | -Quality of newly enrolled students<br>-Quality of students<br>-Quality of graduates<br>-The quality of the researchers<br>-Quality of academic staff | -Study conditions<br>-Research conditions | -Publications with a coauthor from abroad<br>-Citation of researchers and publications<br>-Interdisciplinary publications |
| Tasic [75] | -Average graduation time<br>-Employment opportunity<br>-The number of earnings of graduates<br>-Multidisciplinarity of the study program<br>-Delivery of teaching resources<br>-Ability to use the internet<br>-The size of the teaching group<br>-Number of students who have completed doctoral studies<br>-Questionnaires about the quality of teaching<br>-Professional practice<br>-Student research work<br>-Organisation of teaching<br>-Student exchange<br>-Availability of information on the website | -Teachers with PhD<br>-Accessibility of teaching staff | -Laboratories<br>-Classrooms<br>-Computer equipment<br>-Student service | -Delivery of funds for science<br>-Teacher awards and recognitions<br>-Profit from science<br>-Scientific works of teachers<br>-Citation of teachers in scientific journals<br>-Multidisciplinary research work |
| Petrusic [76] | | -Quality of teaching and learning<br>-Teacher/student relationship<br>-Teachers | -Quality mechanisms of the teaching and research process | -Prestige (visibility)<br>-Citation<br>-Scientific productivity<br>-Excellence in research<br>-IF journal factor<br>-Collaboration indicators<br>-Number of scientific projects |

| Ranking Systems | Indicators | | | |
| --- | --- | --- | --- | --- |
| | **Knowledge Transfer** | **International Orientation/Internationalization** | **Regional Engagement** | **Productivity** |

**Table 4.** *Cont.*

| | | | |
|---|---|---|---|
| HEQAM | | -Career prospects<br>-Connections of institutions with business<br>-Improve technical skills<br>-Improve communication skills<br>-Language skills<br>-Employment opportunities through daycare programs<br>-Opportunities to continue to study abroad<br>-Availability of exchange programs with other institutes.<br>-Opportunity for graduate programs. | |
| Luzanin [74] | -University–industrial research<br>-Finishing works in cooperation with the economy<br>-Student practice | -Academic staff with a doctorate at another domestic or foreign institution<br>-Academic staff who have been engaged in teaching or scientific work abroad (for the last ten years) for at least three months | |
| Tasic [75] | -Encouraging cooperation with the industry<br><br>-Joint research projects with industry<br>-Company training and courses<br>-Work experience of teachers in the industry<br><br>-Joint scientific papers with industry<br>-Companies founded by the Faculty<br>-Patents<br>-Patents with industry<br>-Profits from cooperation with the economy<br>-Earnings from licenses sold<br>-Citation of scientific papers in patents | -Possibility to study for foreign students<br>-Joint study programs with foreign faculties<br>-Networking with foreign faculties<br>-The interest of international students to enrol in college<br>-Profit from international research projects<br>-International projects with foreign faculties<br>-Visiting professors from abroad<br>-Number of international students who have completed doctoral studies at the Faculty<br>-Teachers' scientific papers with colleagues from abroad<br>-Employment in international companies | -Patents with regional companies<br>-Courses and training for citizens<br>-Summer schools for high school students<br>-Final work of students in cooperation with regional companies<br>-Open lectures for all people<br>-Regional companies founded by the Faculty<br>-The interest of high school students from the region to enrol in the Faculty<br>-Earnings from regional companies<br>-Professional practice in regional companies<br>-Employment in regional companies<br>-Scientific papers with regional companies<br>-Research projects with the region |
| Petrusic [76] | | -Student mobility<br>-Teacher mobility<br>-Internationalisation | -Industry revenue<br>-Transmission of research results to society<br>-Career and relevance to the market |

Several national documents in Serbia, e.g., "Education Development Strategy in Serbia until 2020" and "Action Plan for Implementation of the Strategy for Development of Education in Serbia by 2020", are focused on:

- The introduction of quality indicators in HE. This particular action has the following implementation steps (according to the Strategy and Action plan): (1) Definition of a set of indicators for monitoring of the condition of HE; (2) Improvement of accreditation standards; (3) Development of a model for the implementation of indicators (information system).
- Academic studies (bachelor and master)—Introduction of the ranking of study programs with implementation steps (1) Definition of a set of indicators; (2) Analysis of different ranking programs (based on the opinion of employers as well as based on knowledge of students); (3) Systematic inclusion of employers in the procedure of evaluation and ranking; (4) Establishment of a manual for ranking study programs.

Some researchers discuss ranking systems and evaluate HEIs [70,77–81] with positive and negative reactions. The European University Association states that ranking systems and evaluation are increasing in importance. Rauhvargers [80] states that it is vital for universities to be transparent and measure results from the perspective of different stakeholders.

The analysis of these systems implication points to some similarities and differences at the international and national levels. International systems are based on a lesser number of indicators, and they are focused on fewer groups of indicators. National systems usually extend some of the international models with national specifications. Researchers generally base their work on many indicators and a more significant number of dimensions (groups of indicators).

It is also possible to conclude that all of the mentioned systems do not include all (or at least several) stakeholders. Additionally, the mentioned systems do not present weight differences in some indicators or groups of indicators. In this research, the authors present a model with dimensions and groups of indicators focused on stakeholders. The authors introduce a model containing the weight factors of the indicators and provide a mathematical framework to help coordinate specific indicators. This approach is developed to provide a tool for universities and institutions to estimate their quality and rank, and a benchmarking tool to improve their indicators and quality of processes and outcomes.

## 4. Research Methodology and a Set of Indicators Definition

### 4.1. Research Methodology

The methodology of this study on HEIs in Serbia was developed based on a survey questionnaire provided to recognised stakeholders to define KPIs. Statistical methods were used to review and analyse the results of the data obtained from the conducted survey.

The developed questionnaire was sent to the e-mail addresses of the representatives of all accredited HEIs in the Republic of Serbia from the techno-technological (TT) field, and it has not been applied before.

The questionnaire's validity was established using a panel of experts employed in the Republic of Serbian Ministry of Education, Science and Technological Development, who have significant HEIs management experience.

The distribution of the number of accredited HEIs according to the type of institutions from the TT field to whom the questionnaire was forwarded is as follows: 10 universities, 46 faculties and 26 higher technical schools. The distribution of the responses received is as follows: 8 questionnaires from universities, 38 questionnaires from faculties and 19 questionnaires from higher technical schools. Consequently, the response rate was about 79% in total.

The methodology was developed for the Republic of Serbia's Ministry of Education, Science and Technological Development needs and it is implemented as a pilot project, and this is why a significant number of HEIs responded. The importance of methodology reflects in the fact that there is a need for a methodology that will better take into account the needs of different stakeholders, be more balanced and better suited to regional, medium

and small universities since the existing methodologies favour the largest HEIs. In this way, the limitations of the existing methodologies for evaluating and ranking universities (discussed in the introductory part) could be overcome.

The classification of the collected data and their entry into databases is also one of the phases of the statistical method. After defining the variables, the database data are imported into the statistical data processing program (IBM SPSS v.21).

The following methods used in the research during the preparation of this paper are a comparative analysis from domestic and foreign literature, quality engineering methods, descriptive analysis of the business environment and the influencing environmental factors on HEIs, and methods of genetic algorithms (GA).

### 4.2. Set of Indicators Definition

HE indicators are measurable values that educational institutions use to track their progress towards specific business goals. Indicators help educational institutions to monitor and evaluate performance and guide them towards established goals.

According to the needs and requirements of stakeholders (students, parents, employers, state, society), and based on international analysis (ARWY, MULTIRANKING, LEIDEN RANKING, WEBOMETRICS...), national analysis (PERSPEKTYWY, MACEDONIA HEIs...), and research models of quality assessment and ranking of HEIs [19–21], and the international project PESHES, Strategy for the Development of Education in Serbia until 2020+, and other literature analysis [71,72,77–81], a set of indicators will be presented divided into six groups or dimensions (Table 5), namely: Institution, which contains nine indicators; Teaching, five indicators; Science, six indicators; Service users (parents, students), four indicators; Employers/Economy, three indicators; Country/Society, three indicators, which is a total of 30 indicators.

**Table 5.** Proposed indicators.

| ID No. | Description | Calculation | Legend |
|---|---|---|---|
| | | Institution | |
| ID 1 | An average grade from previous educational level | $b = \frac{x}{5}$; | x—an average of enrolled students<br>five—maximum grade |
| ID 2 | Total number of students in the 1st year | $c = n$; | n—number of students |
| ID 3 | % of maximal 60 European Credit Transfer and Accumulation System (ECTS) from previous year | $d = \frac{x}{y}$; | x—number of students with 60 ECTS<br>y—total number of students |
| ID 4 | Number of foreign students | $e = \frac{x}{y}$; | x—number of students<br>y—number of foreign students |
| ID 5 | Percentage of graduates | $f = \frac{x}{y}$; | x—number of graduated students<br>y—number of students enrolled in the first year |
| ID 6 | Finances of HEI (total income) | $i = \frac{x}{y}$; | x—total income<br>y—number of employees |
| ID 7 | Financing of science | $j = \frac{x}{y}$; | x—financing for science<br>y—total income |
| ID 8 | Income from students' fees | $k = \frac{x}{y}$; | x—income from students<br>y—total income |
| | | Teaching | |
| TD1 | Number of study programs | $l = n$; | n—number of study programs |
| TD2 | Number of students in lecturing groups | $m = \frac{x}{y}$; | x—number of students in the group<br>y—number of students at the study program |
| TD3 | Evaluation of study program (students evaluation) | $n = n$; | n—students satisfaction with the study program |
| TD4 | Evaluation of teaching process program (students evaluation) | $o = n$; | n—evaluation of teaching |
| TD5 | Student internship | $p = \frac{x}{y}$; | x—number of students with internship<br>y—total number of students |
| | | Science | |
| SD1 | Published manuscripts | $q = \frac{x}{y}$; | x—number of publications in the last year<br>y—number of research staff |
| SD2 | Number of publication at SCI, SSCI | $r = \frac{x}{y}$; | x—number of publications at SCI<br>y—total number of publications |

**Table 5.** *Cont.*

| ID No. | Description | Calculation | Legend |
|---|---|---|---|
| | | Institution | |
| SD3 | Number of books | $s = \frac{x}{y};$ | x—number of books during the year<br>y—total number of teachers |
| SD4 | Academic staff mobility | $t = \frac{x}{y};$ | x—number of staff with mobility<br>y—total number of staff |
| SD5 | Students' mobility | $u = \frac{x}{y};$ | x—number of students at HEIs abroad<br>y—total number of students |
| SD6 | Publications in international cooperation | $v = \frac{x}{y};$ | x—number of publications with international cooperation<br>y—total number of publications |
| | | Stakeholders (students, parents) | |
| SSD1 | The average duration of studies | $w = n;$ | n—average years |
| SSD2 | Learning outcomes in graduate students | $lo = \frac{x}{y};$ | x—number of students with eight average and higher<br>y—total number of students |
| SSD3 | Unemployment rate of graduates | $up = \frac{x}{y};$ | x—number of graduate students employed during the first year after graduation<br>y—number of students at year |
| SSD4 | Students' scholarship (from industry) | $z = \frac{x}{y};$ | x—number of students with scholarship<br>y—total number of students |
| | | Employers (business) | |
| ED1 | Projects with business entities | $aa = \frac{x}{y};$ | x—number of projects with business<br>y—total number of projects |
| ED2 | Number of BSc and MSc realised with business | $bb = \frac{x}{y};$ | x—number of BSc and MSc realised with business<br>y—total number of BSc and MSc |
| ED3 | Scientific manuscripts with business | $cc = \frac{x}{y};$ | x—number of manuscripts with business<br>y—total number of manuscripts |
| | | Society/State | |
| SS1 | Participation in national projects | $dd = \frac{x}{y};$ | x—number of national projects<br>y—total number of projects |
| SS2 | Number of projects financed by the state | $ee = \frac{x}{y};$ | x—number of projects financed by the state<br>y—total number of projects |
| SS3 | Public lectures | $ff = n;$ | n—number of public lectures |

The authors introduced the following indicators presented in specific defined dimensions:

## 5. Definition and Testing of the Mathematical Model for Ranking

To test the model, the authors took data from the University of Kragujevac, Faculty of Engineering. Data are presented in Table 6.

**Table 6.** KPIs metrics.

| No of Group | Group/Dimension | No | Indicator | ID | Value |
|---|---|---|---|---|---|
| I | Institution | 1 | An average grade from previous educational level | $I_1$ | 0.826 |
| | | 2 | Total number of students in the 1st year | $I_2$ | 303 |
| | | 3 | % of maximal 60 ECTS from previous year | $I_3$ | 0.284 |
| | | 4 | Number of foreign students | $I_4$ | 0.008 |
| | | 5 | Percentage of graduates | $I_5$ | 0.766 |
| | | 6 | Condition of studies (students evaluation) | $I_6$ | 4.33 |
| | | 7 | Finances of HEI (total income) | $I_7$ | 2,549,185.710 |
| | | 8 | Financing of science | $I_8$ | 0.2192 |
| | | 9 | Income from students' fees | $I_9$ | 227,847.222 |
| II | Teaching | 1 | Number of study programs | $II_1$ | 10 |
| | | 2 | Number of students in lecturing groups | $II_2$ | 0.75 |
| | | 3 | Evaluation of study program (students evaluation) | $II_3$ | 4.27 |
| | | 4 | Evaluation of teaching process program (students evaluation) | $II_4$ | 4.69 |
| | | 5 | Student internship | $II_5$ | 1 |

**Table 6.** *Cont.*

| No of Group | Group/Dimension | No | Indicator | ID | Value |
|---|---|---|---|---|---|
| III | Science | 1 | Published manuscripts | III$_1$ | 1.113 |
| | | 2 | Number of publication at SCI, SSCI | III$_2$ | 0.125 |
| | | 3 | Number of books | III$_3$ | 0.0435 |
| | | 4 | Academic staff mobility | III$_4$ | 0.0174 |
| | | 5 | Students' mobility | III$_5$ | 0.0181 |
| | | 6 | Publications in international cooperation | III$_6$ | 0.3047 |
| IV | Stakeholders (students, parents) | 1 | The average duration of studies | IV$_1$ | 4.2 |
| | | 2 | Learning outcomes in graduate students | IV$_2$ | 0.1803 |
| | | 3 | An unemployment rate of graduates | IV$_3$ | 0.6544 |
| | | 4 | Students' scholarship (from industry) | IV$_4$ | 0.0083 |
| V | Employers (business entities) | 1 | Projects with business entities | V$_1$ | 0.3509 |
| | | 2 | Number of BSc and MSc realised with business | V$_2$ | 0.4264 |
| | | 3 | Scientific manuscripts with business | V$_3$ | 0.0859 |
| VI | Society/State | 1 | Participation in national projects | VI$_1$ | 0.2105 |
| | | 2 | Number of projects financed by the state | VI$_2$ | 0.2105 |
| | | 3 | Public lectures | VI$_3$ | 10 |

*Development of the Mathematical Model*

Step 1. KPI $i$, $i = 1, \ldots, I$, values at the level of each HEI $h$, $h = 1, \ldots, H$ and KPI group $g$, $g = 1, \ldots, G$ are obtained by HEIs management teams. These values are denoted as $v_{gi}^h$.

Step 2. Determine the maximal $v_{gi}^{max}$, median $v_{gi}^{mid}$ and minimal $v_{gi}^{min}$ values of each defined KPI $i$, $i = 1, \ldots, I$, within a group $g$, $g = 1, \ldots, G$, at the level of all observed HEIs.

Step 3. Obtained normalised KPI $n_{gi}^h$ $i$, $i = 1, \ldots, I$, $g = 1, \ldots, G$ values at each HEI level, $h = 1, \ldots, H$, calculated within the interval {0, 1}: $n_{gi}^h = \frac{v_{gi}^h - v_{gi}^{min}}{v_{gi}^{max} - v_{gi}^{min}}$.

Step 4. Optimisation functions for ranking of all KPIs, within each group $g$, $g = 1, \ldots, G$, are:

$$\max(S_{total}^{gi}) = \sum_{i=1}^{I} r_{gi} \cdot \sum_{h=1}^{H} n_{gi}^h \tag{1}$$

$$\min(Var_{gi}) = \frac{\sqrt{\sum_{i=1}^{I} r_{gi} \cdot \sum_{h=1}^{H} \left( n_{gi}^h - \overline{n} \right)^2}}{H \cdot I} \tag{2}$$

$S_{total}^i$ is the sum of KPIs rank coefficient values at the level of all KPIs within a group $g$, $g = 1, \ldots, G$,

$Var_{gi}$ is the ranks coefficient variance value at the level of all KPIs within a group $g$, $g = 1, \ldots, G$.

Step 5. By using GA, KPIs rank coefficient values, at the level of each group $g$, $g = 1, \ldots, G$, with respect of all HEIs, are determined.

Step 6. KPIs rank coefficient values are sorted in decreasing order. The rank of the KPIs corresponds to the KPIs' rank coefficient values.

Step 7. Optimisation functions for ranking of all observed HEIs are:

$$\max(S_{total}^h) = \sum_{h=1}^{H} r_h \cdot \sum_{g=1}^{G} \sum_{i=1}^{I} n_{gi}^h \tag{3}$$

$$\min(Var_h) = \frac{\sqrt{\sum_{h=1}^{H} r_h \cdot \sum_{g=1}^{G} \sum_{i=1}^{I} \left( n_{gi}^h - \overline{n_g} \right)^2}}{H \cdot I} \tag{4}$$

where: $S_{total}^h$ is the sum of KPIs' rank coefficient values at the level of all HEIs $h$, $h = 1, \ldots, H$ and group $g$, $g = 1, \ldots, G$,

$Var_h$ is the rank coefficient values variance at the level of all HEIs $h$, $h = 1, \dots, H$ and group $g$, $g = 1, \dots, G$.

Step 8. By using GA, HEIs rank coefficient values concerning all KPIs are determined.

Step 9. For each group of KPIs $g$, $g = 1, \dots, G$, KPI values are aggregated $s_g$ to determine the overall ranking of all HEIs.

$$s_g^h = \left( \sum_{i=1}^{I} n_{gi}^h \right) \tag{5}$$

Step 10. Based on the determined aggregated results $s_g^h$, management initiatives have to be taken to increase the effectiveness and reach the first HEI in the ranking.

According to the obtained data and the presented algorithm (Step 1 to Step 5), the following figures (Figures 2–4) show the ranking of the KPIs within each KPI group (Institution, Teaching, Science, Service users, Employees/Economic, State/Society). Figure 2a represents Institution, while Figure 2b represents Teaching group KPIs ranking. Figure 3a represents Science, while Figure 3b represents Service Users' group KPIs ranking. Finally, Figure 4a represents Employees/Economic, while Figure 4b represents State/Society group KPIs ranking.

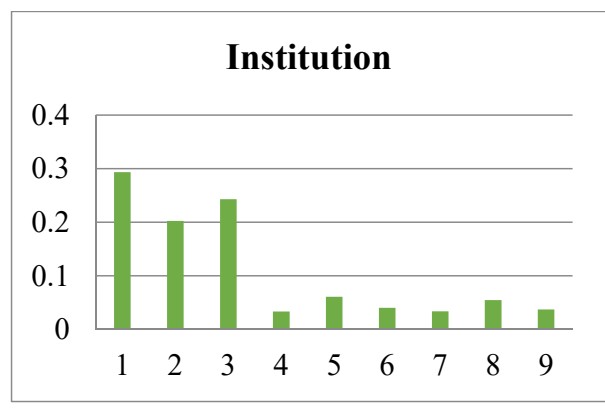

(**a**) Institution KPI group of KPIs

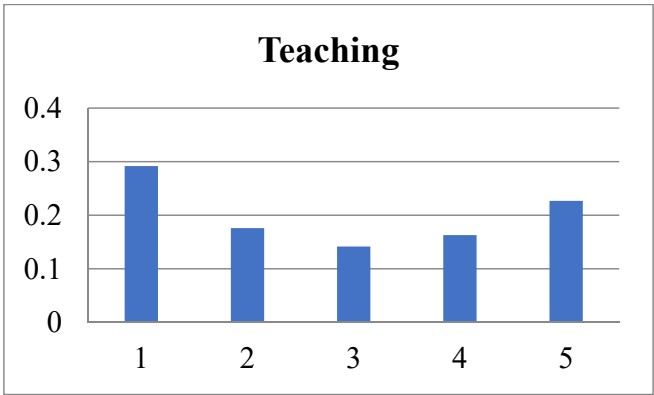

(**b**) Teaching KPI group of KPIs

**Figure 2.** Ranking of KPIs within the (**a**) Institution and (**b**) Teaching group of indices.

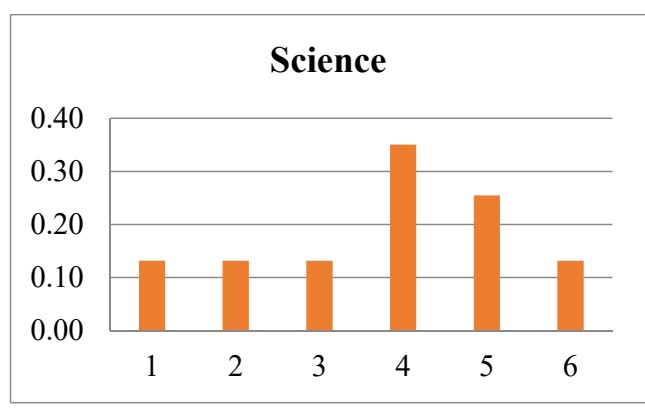

(**a**) Science KPI group of KPIs

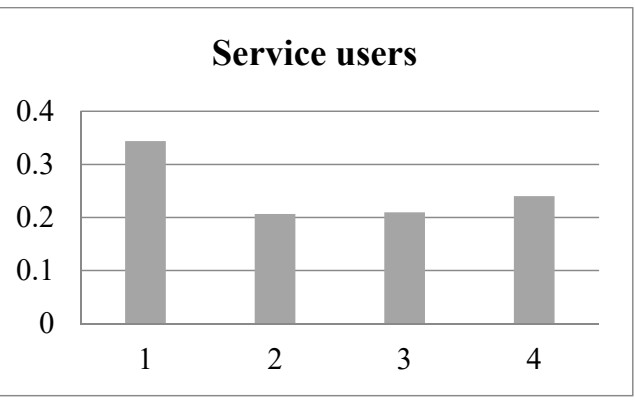

(**b**) Service Users KPI group of KPIs

**Figure 3.** Ranking of KPIs within (**a**) Science and (**b**) Service Users group of indices.

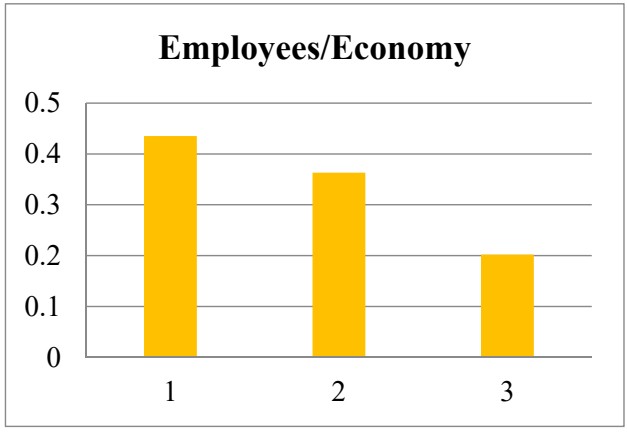

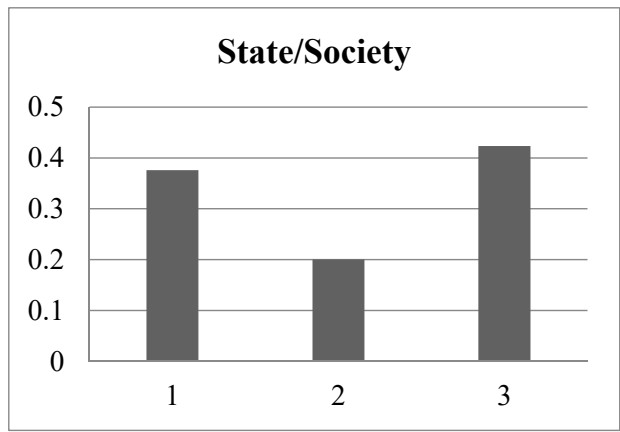

(**a**) Employees/Economy group of KPIs  (**b**) State/Society group of KPIs

**Figure 4.** Ranking of KPIs within (**a**) Employees/Economy and (**b**) State/Society group of indices.

The Pareto front for ranking of HEIs has been obtained (Figure 5) according to the proposed algorithm (step 6 to step 10).

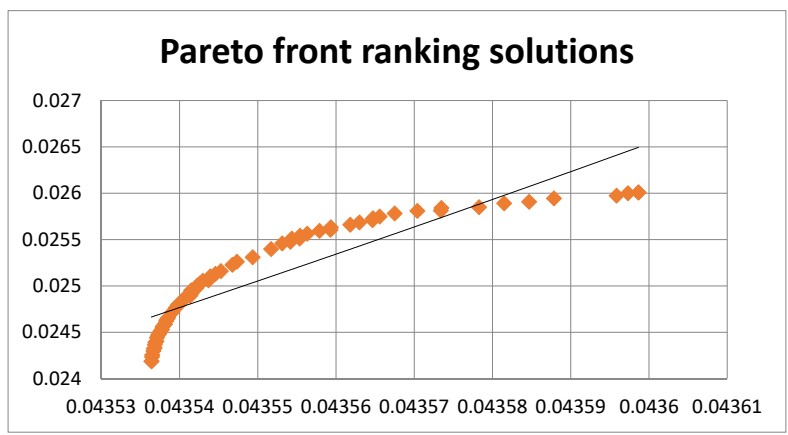

**Figure 5.** Pareto front of the HEIs ranking.

Based on the Pareto front values, the HEI at the first place in the ranking may be determined. Now, the KPI values of the best-ranked HEI can be compared with the values calculated in the second step of the algorithm (Figure 6).

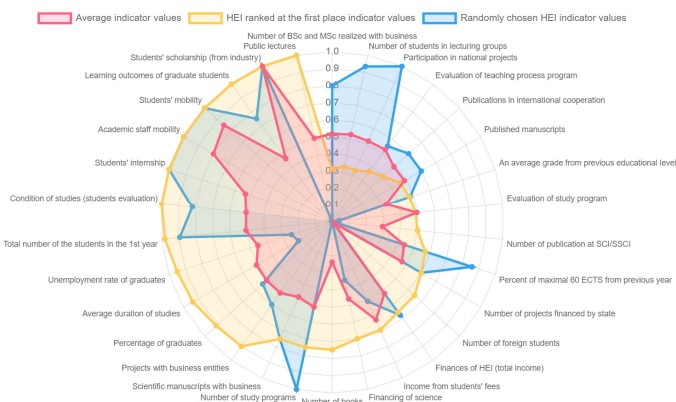

**Figure 6.** Comparison of values between best ranged HEI and values obtained in the second step of the proposed algorithm.

## 6. Scientometrics

While ranking universities, the authors needed to analyse the publication activities of each university in detail. Thanks to digital technologies such as InCites, it is possible to directly see the publication activities of the University of Kragujevac. These technologies will help analyse and develop the tools to enhance the research details of the university [81]. Accordingly, Figure 7 shows that the number of Q1 and Q2 publications has decreased sharply by 18% since 2017—from 142 to 117.

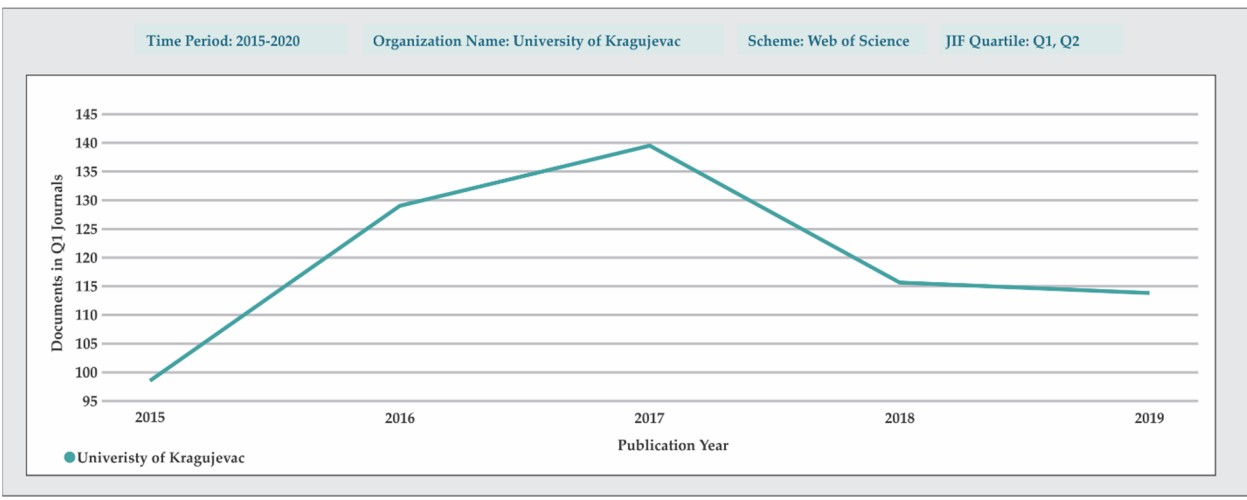

**Figure 7.** The number of Q1 and Q2 publications of the University of Kragujevac.

If we filter the Research Area by Engineering only, we find out that the number of Q1 and Q2 publications was reduced by 36% (from 22 articles to 14) (Figure 8).

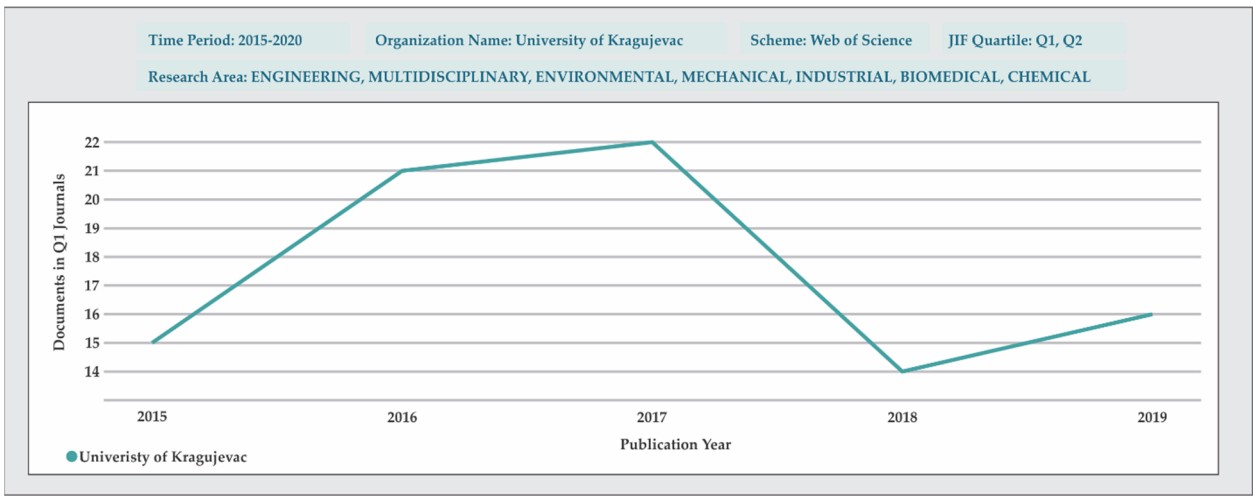

**Figure 8.** The number of Q1 and Q2 publications in Engineering—University of Kragujevac.

The current digitalisation will help us to find the reasons for the decline in publication. First of all, the authors suggest analysing the number of articles from each of the research staff per year (Figure 9).

These results will allow us to define the growth or loss of each scientific group. The same method can be used to estimate the trends of grants and collaborations. As a result, the authors can see which groups of scientists have reduced the quality or quantity of their publications. The authors then develop the necessary tools to avoid the problem of reducing the number of high ranking articles.

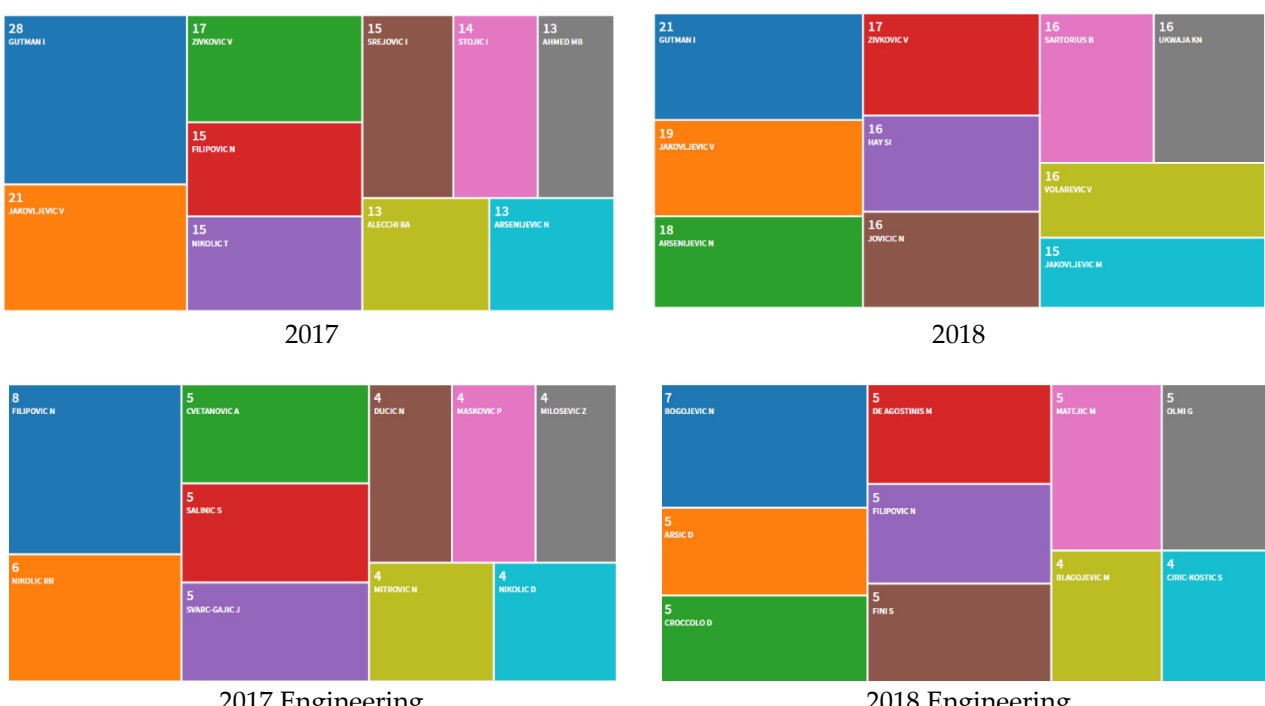

**Figure 9.** The number of articles about research staff per year.

## 7. Discussion

The analysed ranking methods represent one of the criteria according to which the authors of this paper selected and classified specific research indicators. The paper [63] contributes to creating a ranking list generation that improves the current situation in methodological and substantive bases with two contributions: integrating new information and using new ranking techniques. The main criticisms are monodimensionality, statistical robustness, dependence on university size and subject mix and lack of consideration. The authors [82] conducted a systematic review of university ranking systems to evaluate the usefulness of the ranking systems and identify opportunities to support and improve research quality and performance. They concluded that future research should evaluate three research outcomes: scientific impact, economic outcomes and public health impact. The author of [83] presented a developed performance-based method for a central planner to allocate research funding to different universities to stimulate the research output. Although this paper focuses on China's HE system, the research framework is general and can be applied to other countries. Each of the mentioned systems looks at the HEIs' individual quality segments differently and evaluates them differently. These facts indicate that the issues reflect the vaguely defined criteria according to which the indicators should be selected. According to all the above, it is clear that there is a need for an approach that will include attitudes about quality in the academic community by the students and employers, government institutions, and others. The mentioned ranking methodologies represent a significant starting point for further research within this paper.

The presented model for quality assessment and performance assessment (based on KPIs) of study programs could be used to measure, monitor and improve HEIs and the education system quality. It could also be an essential tool for benchmarking, decision support and finally, a step towards financing educational systems according to their performances. The results for all the considered KPI groups divided by dimensions (Institution, Teaching, Science, Service Users, Employers/Economic, Country/Society) were obtained by applying the presented solution. Figure 2a shows the ranks of KPIs within the Institution group. The Institution group contains nine KPIs, where the most influential is the indicator $KPI_{ID1}$ (an average grade from previous educational level), then the indicator $KPI_{ID3}$ (% of maximal 60 ECTS from the previous year), and the indicator $KPI_{ID2}$ (total number of students in

the 1st year). These three indicators are ranked the highest compared to the remaining six indicators in this group. The indicators $KPI_{ID9}$ (Income from students), the indicator $KPI_{ID4}$ (number of foreign students), and the indicator $KPI_{ID7}$ (financing of science) have the lowest rank, which is the least important from the quality aspect, but undoubtedly crucial from the HEIs business aspect. The indicators $KPI_{ID5}$ (Percentage of graduates), $KPI_{ID6}$ (Finances of HEI (total income)f), and $KPI_{ID8}$ (Income from students' fees) are of medium importance. These results show that the previous educational level quality plays an essential role in HIEs current quality performance.

Based on the presentation in Figure 2b, in the group Teaching, it can be concluded that the $KPI_{TD1}$ (Number of study programs) has the highest value or rank, while $KPI_{TD3}$ (Evaluation of study program (students evaluation)) has the lowest rank. Other indicators, $KPI_{TD2}$ (Number of students in lecturing groups), $KPI_{TD4}$ (Evaluation of teaching process program (students evaluation)), and $KPI_{TD5}$ (Student internship), have average values that are half as significant as the highest-ranked $KPI_{TD1}$. These results show that the study programs diversity is a significant factor in HEIs quality determination.

Figure 3a presents a range of KPIs within the Science group, which contains six indicators with their values. The values of $KPI_{SD4}$ (Academic staff mobility) represent the highest-ranked indicator, while the indicator $KPI_{SD5}$ (Students' mobility) has a slightly lower value and rank. The remaining four indicators, $KPI_{SD1}$ (Published manuscripts), $KPI_{SD2}$ (Number of publication at SCI, SSCI), $KPI_{SD3}$ (Number of books), and $KPI_{SD6}$ (Publications in international cooperation), have equal values and the lowest rank compared to higher-ranked KPIs. Although they have a lower rank and not so much impact on the quality of HE, they are significant since they can be applied to assess teaching staff quality. These results show that mobility for both academic staff and students is essential since it leads to the exchange of knowledge and culture at the international level.

The Stakeholders group (Figure 3b) contains four indicators, of which the $KPI_{SSD1}$ (The average duration of studies) has the highest ranking in the group, while the lowest-ranked indicator is $KPI_{SSD2}$ (Learning outcomes in graduate students). The indicator $KPI_{SSD3}$ (The unemployment rate of graduates) and the indicator $KPI_{SSD4}$ (Students scholarship (from industry)) are in the middle compared to other KPIs in the group. These results show that the average duration of studies is essential since the labour market needs to be provided with a trained workforce in the short term.

The rank of KPIs within the group Employers/Business is presented in Figure 4a. The highest rank has $KPI_{ED1}$ (Projects with business entities). The $KPI_{ED2}$ (Number of BSc and MSc realised with business) is ranked lower, while the $KPI_{ED3}$ (Scientific manuscripts with business) is ranked the lowest. These results show that the most critical performance is related to the cooperation with business entities since such cooperation produces new research ideas, which can contribute to the sustainable development of both HEIs and business entities.

The ranking of KPIs within the State/Society group is presented in Figure 4b. The $KPI_{SS3}$ (Public lectures) is ranked highest, the $KPI_{SS1}$ (Participation in national projects) is of medium importance, while the $KPI_{SS2}$ (Number of projects financed by the state) has the lowest rank. These results show that public lectures have a significant impact since they lead to an improved public image of HEIs.

Based on the Pareto front values (Figure 5), it can be determined which HEI is first in rank. The values of the best ranked KPIs of the HEIs were compared with the values determined in Step 2 of the proposed algorithm. Figure 6 presents a comparison of the values of the KPIs between the best ranked HEIs, randomly selected HEI values of the KPIs, and average values of the KPIs at the level of all considered HEIs. By optimising specific KPIs, the performance of the HE process can be improved. Improving KPIs will ensure a better ranking of HEIs and improve performance in terms of quality. The main advantage of this model compared to the existing evaluation model is manifold: it has more dimensions, a novel mathematical approach, and is more suitable for general HEIs.

## 8. Conclusions

*8.1. Main Findings*

The definition of the model for performance-based evaluation in HE (institutions and study programs) has considerable importance for different target groups:

(1) The academic community and the universities in a specific country. Developing a system for measuring performance and multidimensional ranking of study programs and institutions should contribute to a better, higher quality, more efficient, market-oriented and socially responsible management of study programs and universities. Through KPIs, the ranking of study programs and institutions will enable focusing on critical processes, process benchmarking, comparison and thus improvement of key processes at universities in Serbia, which should contribute to higher levels of education, research, development and innovation processes, broader internationalisation as well as cooperation with industry both locally and across the region. On the other hand, all this affects the definition and redefines institutional strategies.

(2) Students. According to the ranking of study programs and institutions, students will have opportunities to make choices that suit them best, bearing in mind the set of performance indicators which would indicate the essential parameters (for example, the number of unemployed graduates in a study program).

(3) Industry and business. Business entities would have an overview of different orientations and parameters of defined study programs, market orientation and quality. In this way, it is possible to achieve feedback between HEIs and industry.

(4) The National Employment Service and the Statistical Office of the Republic of Serbia's focus groups would benefit from the access to real data, their organised monitoring and better connections with HEIs and businesses. Today, for example, it is not possible to generate information on the number of unemployed graduates originated from individual institutions,

(5) Government and state institutions (policymakers) can use the defined set of performance criteria to create the legal framework and recommendations for funding or financial models that should be incorporated as indicators for measuring the quality of the educational process.

This paper aims at defining and presenting a model for assessing the quality of HEIs in the technical–technological field and the decision-making system to support and adopt optimal management strategies for quality improvement.

There were two primary scientific objectives. The first objective was to develop a model for assessing the quality of HEIs in the technical–technological field from the aspects of various stakeholders. The first objective was to develop a system for decision support and making optimal decisions to improve individual quality indicators to define appropriate management strategies to improve quality. A model of KPIs was presented, i.e., a mathematical model was created that enabled the assessment of the quality of study programs and HEIs from the perspective of different stakeholders and the ranking and comparison of study programs and HEIs.

Quality in the HE sector is one of the current issues in the academic community. When monitoring and researching the HE field, analysing the literature and the current situation in the system of HE in developing countries, it can be concluded that there is no single way to assess the quality of HEIs. This knowledge was a good starting point for the research presented in this paper. Accordingly, the findings include developing a system for the quality assessment and ranking of HEIs. Additionally, evaluating the relevance of the KPIs of HEIs differs from the viewpoints of the different stakeholders. However, it is possible to develop a system for decision support and selection of the optimal strategy for improving the performance of study programs and HEIs from the quality aspect.

The realisation of more comprehensive and specific objectives needs to provide public value management, moving from bureaucracy to a market approach while enabling HE to serve society better.

Accordingly, H1 (It is possible to develop a system for quality assessment and ranking of higher education institutions) and H2 (It is possible to develop a system to support decision-making and selection of the optimal strategy for improving the performance of study programs and higher education institutions from the aspect of quality) have been proven.

The presented model has several advantages. In the first place, this model introduces different perspectives (these perspectives are different from perspectives in existing models). Secondly, this model introduces weights for all KPIs. Finally, this model enables ranking and benchmarking between institutions. This model considers all stakeholder views and the number of indicators from a specific environment, so it is more friendly and accurate for smaller and regional universities. Finally, we introduced a software solution that enables HEIs to monitor, track, and improve their indicators and essential processes. The suggested model has great potential because the more universities use this model, the broader and more useful benchmarking will be.

### 8.2. Theoretical Implications

The theoretical implications reflect the defined model for quality assessment and performance assessment (based on KPIs) of study programs and HEIs while considering a complex group of stakeholder requirements. The model has the following novelty: the definition of specific perspectives, a unique set of KPIs (respecting demands of all stakeholders), and the introduction of weights for specific KPIs. The presented mathematical model with nine steps using GA optimisation is an essential contribution of this manuscript. This model opens up the way to define the methodology for measuring, monitoring and improving the HEIs and the education system quality.

As there was no clearly defined methodological approach that combined the application of business intelligence tools for HEIs process and performance management, the paper has theoretical implications which are reflected in:

(1) Defining models to support decision-making on quality objectives and business performance in HEIs;

(2) Determining the measure of execution of processes, subprocesses and their KPIs based on the realised results in the form of performance for small and medium HEIs;

(3) Determining and optimising specific KPIs based on the observed performance that needs to be improved and;

(4) Predicting HEIs' performance improvements based on established optimal KPI improvements.

### 8.3. Practical Implications

The practical implications include defining a decision support system that will enable the adoption of optimal decisions by the HEI's management team to improve the performance of study programs and the HEI. The model enables identifying the most influential KPIs of HEIs currently, so managers and decision-makers are allowed to select KPIs for optimal improvement. By determining the optimal improvements, it is possible to identify and solve business problems by applying an appropriate business strategy and improving overall HEIs' performance.

The presented system may enable benchmarking. It enables each HEI to compare their processes or subprocesses with the rank of processes or subprocesses of all other HEIs based on the achieved performance and the quality of the analysed service processes. This solution makes it possible for HEIs to quantify the measure of achieving the set business goals.

Additionally, the solution enables the simulation and verification of different scenarios for improving the quality and performance of the system, which can have highly positive effects on the management and improvement of higher education.

Consequently, it may be concluded that the practical implications of an integrated ranking solution are reflected in providing opportunities for organisations to report and identify bottlenecks and influencing factors, solve dynamic HEIs' problems, monitor

HEIs' processes, learn from leading HEIs, and compare processes and subprocesses in different HEIs.

The originality of the research lies in the presented model that can be made available to government institutions and serve as a basis in the overall ranking and evaluation of HEIs with the possibility to develop a performance-based funding system. Additionally, other stakeholders can have an insight into the performance of an institution for the sake of their own needs and goals.

### 8.4. Study Limitations

Within the research, certain limitations are primarily reflected in the selection and number of HEIs from the technical–technological field, which raises issues and questions about whether the presented model applies to all HEIs in all academic areas. Furthermore, the limitations are primarily related to selecting specific HEIs (small and medium-sized), so the question is whether the model applies to large HEIs. The model was tested on data obtained from organisations in the Republic of Serbia, so the results are not valid for decision-making in HEIs outside its borders.

In addition to the abovementioned, there are also limitations related to the validity of the data obtained from HEIs. The data were obtained from the HEIs' managers and decision-makers, so there is a risk that the obtained data are not objective in reviewing the estimated values of processes, subprocesses, KPIs and HEIs' performance.

### 8.5. Future Lines of Research

Directions for further research should be focused on covering all HEIs (faculties, academies, and colleges) in developing counties, where the defined and proposed model can be implemented. Consequently, the ranking of all HEIs can be carried out at the state level, based on the proposed model, and the national ranking system of HEIs can be suggested.

Further research can employ the concept of innovative education and smart universities, which implies and is based on a new approach to education, learning and education strategy, distance learning, and the use of modern classrooms and laboratories.

Innovative education provides a framework for learning in new circumstances, based on trends and strategies of education and sustainable success.

Preparing students for a new role in the 21st century is the primary goal of intelligent education. This goal requires that the education system, from primary to HE, be redesigned in line with the new demands and challenges of the 21st century.

**Author Contributions:** Conceptualisation, Z.L. and A.Đ.; methodology, Z.L.; software, A.Đ.; validation, A.G. and Z.L.; formal analysis, A.G.; investigation, Z.L.; resources, A.G.; data curation, A.G.; writing—original draft preparation, Z.L.; writing—review and editing, A.Đ.; visualisation, ZL; project administration, A.G.; funding acquisition, A.G. All authors have read and agreed to the published version of the manuscript.

**Funding:** This research was funded by the Ministry of Science and Higher Education of the Russian Federation, grant number 075-15-2020-934, dated 17 November 2020.

**Institutional Review Board Statement:** Not applicable.

**Informed Consent Statement:** Not applicable.

**Acknowledgments:** The research is partially funded by the Ministry of Science and Higher Education of the Russian Federation as part of the World-class Research Center program: Advanced Digital Technologies (contract No. 075-15-2020-934, dated 17 November 2020).

**Conflicts of Interest:** The authors declare no conflict of interest.

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
