# Peer review of "Improvement of Quality of Higher Education Institutions as a Basis for Improvement of Quality of Life"

_sustainability, doi:10.3390/su13084149_

Round 1
Reviewer 1 Report
The strengths of the article is original and interesting considerations with is consistent with the pattern of research. Solid methodology of the research with statistical analysis.
Therefore contribution to existing knowledge is considerable. Also advantage of the research is perfect organization & readability.
I cannot find the weaknesses of the assessed article. Model article worthy of imitation.
In generally it is excellent article and very interesting considerations, which is consistent with the pattern of research. A very good review article with the analysis of statistics on the topic under study.
Overall evaluation: article it is suitable for publication in current version.
Author Response
Comments and Suggestions for Authors
The title is long and not very appealing.
Improvement of quality of higher education institutions as a basis for improvement of quality of life
The Abstract must have the following logic:
Purpose: This paper aims to propose the quality assessment model for higher education institutions from the technical-technological field and the system for decision support and optimal management strategies for quality improvement.
Design /
methodology / approach; Obtaining research results will be based on surveying stakeholders in higher education and obtaining quantitative data regarding the key performance indices. Quantitative data and genetic algorithm method will be applied to determine optimal management strategies for quality improvement.
Findings; Quality in the higher education sector is one of the current issues in the academic community. Monitoring and researching the higher education field, analyzing the literature and the current situation in the system of higher education in developing countries, it can be concluded that there is no single way to assess the quality of higher education institutions. This knowledge was a good starting point for the research presented in this paper. Accordingly, the findings include developing a system for quality assessment and higher education institutions ranking. Also, evaluating the relevance of key performance indicators (KPIs) of higher education institutions differs from different stakeholders' aspects. However, it is possible to develop a system for decision support and selection of the optimal strategy for improving study programs' performance and higher education institutions from the quality aspect.
Practical implications; The practical implications include defining a decision support system that will enable the adoption of optimal decisions by the higher education institution's management team to improve the performance of study programs and the higher education institution. The presented system may enable benchmarking, and simulation and verification of different scenarios for improving the quality and performance of the higher education institutions.
Originality
/ value; The originality of the research lies in the presented model that can be made available to government institutions and serve as a basis in the overall ranking and evaluation of higher education institutions with the possibility to develop a performance-based funding system. Also, other stakeholders can have an insight into the institution's performance for the sake of their needs and goals.
Its introduction is very weak. Lacks bibliography. There are only two
authors cited!
The literature GAP needs to be clarified and reinforced. The purpose
of the study is also unclear. I recommend redoing the entire
introduction:
1- Make a frame for the reader
International practice shows numerous examples of HEIs rankings (Ramasamy et al., 2016; Dai et al., 2016; Nazari-Shirkouhi et al., 2020). However, within these systems, different stakeholders' existence and different needs and expectations towards HEIs in developing, measuring, and monitoring performance systems to assess the quality of HEIs are not entirely considered (Alach, 2016; Adhikari, 2019). For instance, authors (Garde Sanchez et al., 2020) state that the Shanghai ranking list must be considered highly controversial and questionable since it cannot assess the complex issue of HEIs quality and addtioanlly state that every change undertaken by HEIs is usually highly controversial and subject to criticism from the different stakeholders' perspectives. Another vital ranking system is Forbes Magazine America's Best Colleges, and this system introduces a new dimension according to which student success is ranked during and after graduation. This ranking methodology's essence is to emphasize academic institutions' strength through their former students' success in the labour market (Woodall et al., 2014; Roth & McAndrew, 2018) and not through the achievements of scientists themselves within institutions. Therefore, these examples have some shortcomings, which are primarily reflected in the one-dimensional experience of quality and its ranking.
In this context, quality assessment and monitoring through indicators become a necessary management tool. Each of the processes carried out at HEIs has its performance that can be measured through key performance indicators (KPIs) (Epifanić et al., 2021). For this paper's purpose, the authors will present a model for quality assessment and ranking for Serbian HEIs and a decision support system to improve HEIs' performance in the technical and technological field. Such a system could provide some benefits for various stakeholders, starting from the possibility of performance-based financing to the satisfaction of the end-user of educational services (Baporikar, 2021). Considered ranking methodologies represent an essential starting point for further research in the scientific world to improve HEIs ranking systems.
According to (Martin & Sauvageot, 2011; Nurcahyo et al., 2018; Martin, 2018; Rymarzak, M., & Marmot, 2020), developing states oblige HEIs to show better results, specifically when it comes to strengthening their management capacity, information systems and quality monitoring tools. Academic communities are innovating different ranking HEIs methodologies, divided into two groups, based on academic and non-academic criteria. Academic criteria imply the establishment of ranking criteria according to the achievements of HEIs themselves, i.e. achievements for which teachers and researchers are responsible (Gontareva et al., 2019; Agasisti et al., 2019). In contrast, non-academic criteria are focused on current and former university students' achievements, i.e. alumni members' success.
Serbia introduced the accreditation process by signing the Bologna Declaration in 2003, which obliged the signatory countries to approach their educational systems' quality assurance responsibly, following the general principles and guidelines (Fernández-Cano et al., 2018; Franceško et al., 2020; Epifanić et al., 2021) of the Bologna process. In this way, Serbia has made the first and significant step on the path to its HEIs ranking system development (Franceško et al., 2020). There is a need for an approach that will include attitudes about quality, academic community and students and employers, and government institutions.
2- Issues of the topic under analysis
Higher education is the driver of society's development since it drives the development of the economy and society and represents the mainstay of achieving a successful career for each individual. Therefore, this paper's main topic is the quality of higher education, which is seen as an aspiration towards continuous improvement of all HEIs processes and their outcomes in achieving the ideal economy and society based on knowledge (Chou, & Gornitzka, 2014).
By analyzing the literature and the current situation in the higher education system in developing countries, it can be concluded that there is an idea of the importance of study programs, HEIs, and education systems quality at all levels, even in the conditions of a COVID-19 pandemic (García-Peñalvo et al., 2021). Quality assessment of HEIs is a vital issue since it can be used for ranking educational institutions, performance-based funding by the state, defining development strategies for improving HEIs, and complete social and economic development (Kayani, 2017; Sam, 2018). That is why the quality of higher education and HEIs is at the centre of interest in international legislation (Duerrenberger and Warning, 2018). In this paper, the authors will attempt to answer the following research questions: How objective is the ranking conducted by the most crucial ranking lists of universities in the world? Is accreditation sufficient to assess the quality of higher education institutions in the Republic of Serbia? What are the different methodologies and criteria for assessing the quality of higher education?
3- Evidence of the GAP of the literature based on the literature
A comprehensive search and analysis of numerous literature were performed to define quality and appropriate methodology (Owens, 2017). Given that the concept of quality in higher education is not one-dimensional, there is no single definition that would encompass all its connotations. The authors (Gunn, 2018) define quality through the following characteristics: quality as excellence, quality as compliance with standards, quality as a convenience, quality as value for money and quality as transformation. Each group of education beneficiaries and stakeholders, i.e. students, parents, academia, employers, and the state, has a different perspective of quality. For example, students associate quality with their institution, the chosen study program, the module, and their diploma title after graduation. Students want to ensure an advantage in the labour market concerning the competition with their degree of education and achieve quality as excellence (Santos et al., 2020). Employers deal with quality in terms of finished products, which are students with acquired competencies for the education system's labour market. They want to hire graduates who possess a high level of knowledge and skills to cope with business challenges and business complexity, hoping to gain an advantage over the competition (Kryscynski et al., 2021). In the developing states, which are the prominent financier of higher education, there is a need for the resources to be used efficiently to achieve a satisfactory level of quality, that is, to achieve quality for the money invested (Ahmed, 2020). In this way, developing states may achieve part of their strategic goals and sustainable development with the help of education systems because as an individual progresses through quality education, the states acquire a quality educated society (Leal Filho et al., 2019). Different views of stakeholders on the quality of higher education are sufficient to define new tasks, and that is to define the criteria based on which the quality of higher education is assessed. Therefore, to define quality in HEIs, all stakeholders should be involved to see the needs, requirements, and expectations of quality from different perspectives. According to the needs, requirements, and expectations of all these stakeholders, the authors of this paper will attempt to define indicators and develop a system for measuring and evaluating HEIs' quality (Choi, 2019).
4- Purpose of the study
The study aims to present a model for assessing the quality of HEIs in the technical-technological field from various stakeholders' aspects. Furthermore, the study aims to present a system for decision supporting and making optimal decisions on improving individual quality indicators to define appropriate management strategies and improve quality.
5- Originality of the study
In this paper, the authors pay attention to selecting the relevant variables and the empirical approaches that can be used to measure HEIs quality. More specifically, the authors will innovate current academic literature in the field by combining different approaches, such as quantitative statistical methods and multiple-criteria optimization, to define and develop the HEIs ranking model. The entire developed model can be placed at the disposal of government institutions and serve as a basis in the ranking and evaluating HEIs to develop a performance-based funding system.
6- What are the expected results (to captivate the reader)
The expected results presented in the paper are theoretical and practical. The expected theoretical result will be reflected in the defined model for quality assessment and performance assessment (based on KPIs) of study programs and HEIs while considering a complex group of stakeholders requirements. This model opens the way to define the methodology for measuring, monitoring and improving the HEIs and the education system quality. The application result includes the definition of a decision support system that will enable optimal decision-making by the management team of the HEIs to improve performance at the level of study programs and HEIs. In addition to the above, it will be possible to conduct benchmarking, i.e. comparing performance with the best in the class and learning from successful institutions. The developed system can enable the simulation and verification of different scenarios for improving the system's quality and performance, which can give highly positive effects on the management and improvement of the given HEIs.
7- The last paragraph should briefly describe what the reader can read
in the following sections.
The paper is organized as follows. The authors introduce quality in HEIs from different stakeholders' perspectives, performance evaluation and assessment issues, and international and national ranking systems in section two. The authors described key performance indicators divided into six dimensions in the third section. While in the fourth section, the metrics, mathematical model, and model application results are introduced. Section five introduces scientometrics as a platform that could be integrated with the presented mathematical model to obtain a comprehensive university's research details. Finally, section six will discuss the results and finish with concluding remarks.
After the introduction, there has to be a literature review.
Academic institutions have recently been hit by significant reforms aimed at improving their performance levels. These reforms are inspired by various factors, such as budget constraints imposed by national governments, HEIs comparison at the global level, and higher education's marketing sector (Krüger et al., 2018).
Through literature analysis in the HEIs quality assurance field, shortcomings have been noticed that are not treated sufficiently. These shortcomings represent the lack of a unified quality assessment and ranking system for HEIs in the European Union countries (Case, Marshall, Linder, 2010). One of the reasons for the lack of a unique quality assessment system may be found in the fact that the quality of a HEIs has different meanings for different groups of education users and stakeholders. This fact is one reason why standard quality assessment models, such as SERVQUAL or other proposed models that are more or less one-dimensional, cannot be applied (Soares et al., 2017). Authors (García-Aracil and Palomares-Montero, 2010) state that HEIs are undergoing essential changes involving the development of new roles and missions, with implications for their structure. They also claim that there is difficulty establishing classification criteria for existing indicators, on which there is no consensus.
Therefore, the need to develop a model for the HEIs quality assessment in the techno-technological field from different stakeholders perspectives has gained attention (Miller, 2016). Although many quality assessment models are developed for the manufacturing and industrial sectors, they cannot be applied directly to the higher education sector. Some previous studies have shown that the manufacturing and higher education sectors' natures are different (Brookes and Becket 2007, Massy et al., 2013). Different models that include performance and their key performance indicators (KPIs) should be developed and tested to meet HEIs' requirements (Rosa et al., 2012; Dick and Tarí, 2013).
Authors (Rosa et al., 2012) define quality as a multidimensional concept depending on the different stakeholders' views resulting in different quality dimensions and performances indices. Due to the vague concept of quality and different meanings for stakeholders and the complicated nature of the educational process and service (Becket & Brookes, 2008), many authors find it challenging to manage higher education quality. Information on the quality of study programs or HEIs, their status compared to other faculties and programs (Altbach, 2010) becomes vital for choosing between many HEIs and study programs (Blanco–Ramirez and Berger, 2014). E-learning becomes a standard in the COVID 19 crisis, so it is essential to understand the impact of e-learning on society and its benefits. The authors (Cidral et al., 2018) aimed to find the determinants of user-perceived satisfaction, use, and personal impact of e-learning. The study proposed a theoretical model integrating theories of information systems' satisfaction and success in the e-learning systems. The model was empirically validated in HEIS through a quantitative structural equation modelling method. The drivers of user-perceived satisfaction are information quality, system quality, instructor attitude toward e-learning, diversity in assessment, and learner perceived interaction with others. System quality, use, and user-perceived satisfaction explain the personal impact. In the study, authors (Al-Rahmi et al., 2018) attempted to mitigate the literature gap concerning E-learning and social media use for active collaborative learning and engagement and its effect on Malaysia's research students' learning performance. The study concludes that overall, active collaborative learning and engagement through social media enriches students' learning activities and facilitates group discussions, and hence, their use should be encouraged in learning and teaching processes in HEIs.
Furthermore, in an educational context characterized by globalization, reputation constitutes a crucial issue for modern HEIs. The investigation results on the relationship between internationalization and reputation in top HEIs reveal that internationalization positively influences a university's reputation and moderates the relationship between its reputation and its institutional performance concerning research quality, teaching quality and graduate employability (Delgado-Márquez et al., 2013).
Consequently, the need to define and evaluate performance indicators in HEIs arose primarily due to the need for measurable and objective quality indicators, which until recently depended only on experts' opinion in the field. It is necessary to include multiple sources of information to provide objective explanations that reflect all higher education complexity in the HEIs quality assessment model (Chalmers, 2008a). The authors (Kanji et al., 1999) examined how TQM principles and core concepts can be measured to provide a means of assessing the quality of different institutions on various aspects of their internal processes. It is found that the quality assurers could use the measurement method in the UK to assess the education quality of HEIs.
Special attention is drawn to applying assessment systems based on KPIs to monitor and measure HEIs' quality. Different types of indicators are distinguished in the literature. The development of a model for quality of HEIs assessment through a KPIs system creates a basis for comparison in the developing states and the world (Findler, 2019). The development and implementation of a system of assessment through KPIs (Kallio et al., G. 2017) could have a broader goal: to improve the management, operation, and quality of HEIs and education systems globally. However, after reviewing the literature, it can be concluded that there is no unique system based on quality KPIs. According to the purpose and area of application in the literature, indicators are defined in different ways. It is clear that there is no universal metric or universal set of KPIs that would enable objective evaluation of the quality and position of HEIs, but it is clear that there are numerous approaches that, to a greater or lesser extent, try to rank HEIs (Mensah, 2020).
Most authors agree (Zwain et al., 2017; Biloshchytskyi et al., 2017; Soewarno & Tjahjadi, 2020) that KPIs can be defined as measures that provide the context of information and statistics, enabling comparisons between different areas alongside other accepted standards. KPIs provide information on how public stakeholders are satisfied with the institution and the entire higher education sector and how education goals have been met within the institutions and the entire higher education sector. KPIs can facilitate sustainable educational policy implementation, inform HEIs' about possible problems, and determine some of the causes of the problems (Fleacă et al., 2018). KPIs are becoming increasingly important since they enable monitoring and assessing various areas' situation to conclude quality objectively (Kooli, 2019). Well-selected and measurable KPIs should be carefully defined and used to determine which parts must take specific actions, i.e. where quality can be improved (Khashab et al., 2020). Quality KPIs represent empirical information that gives a picture of how a HIEs realizes its goals and ensures continuous monitoring of its quality level (Yusof et al., 2018). KPIs should provide information on whether the HEIs have achieved the planned results and achieved the set goals and to what extent they deviate from the planned values. Quality indicators make it possible to monitor performance for comparison, facilitate the assessment of institutional functioning, and provide information for external quality evaluation needs (Chalmers, 2008).
Does the study have no methodology? After the literature review, the
methodology has to come. The methodology has to be detailed. Is the
study quantitative or qualitative? What method is used? Is it a
bibliometric review? This needs to be justified and explained.
This study's methodology on higher education in Serbia was developed based on a survey questionnaire provided to recognized stakeholders to define key performance indicators. Statistical methods were used to review and analyze the results of the data obtained from the conducted survey. Classification of collected data and entry into databases is also one of the phases of the statistical method. After defining the variables, the database's data are imported into the statistical data processing program (IBM SPSS v.21).
The following methods used in the research during this paper's preparation are: a comparative analysis from domestic and foreign literature, quality engineering methods, descriptive analysis of the business environment and the influencing environmental factors on HEIs, and methods of genetic algorithms (GA).
In addition to the above, methods that imply that performance management's issue can be viewed as a multi-criteria decision-making problem are applied.
In discussing the results, the authors do not confront them with the literature.
The paper (Daraio et al., 2015) contributes to creating a ranking list generation that improves the art's current state on methodological and substantive bases with two contributions: integrating new information and using new ranking techniques. The main criticisms are mono dimensionality, statistical robustness, dependence on university size and subject mix and lack of consideration. The paper provides an experiment that deals with these problems regarding universities in Europe. The results provide evidence on how to separate the different performance dimensions to identify a direction for improvement. The authors (Vernon, M. M., et al.,2018) conducted a systematic review of university ranking systems to evaluate ranking systems' usefulness and identify opportunities to support and improve research quality and performance. They concluded that future research should evaluate three research outcomes: scientific impact, economic outcomes and public health impact. The author (Wang, 2019) presented a developed performance-based method for a central planner to allocate research funding to different universities to stimulate the research output. Although this paper focuses on China's higher education system, the research framework is general and can be applied to other countries. Each of the mentioned systems looks at the HEIs individual quality segments differently and, therefore, evaluates them differently. These facts indicate that the issues reflect the vaguely defined criteria according to which the indicators should be selected. According to all the above, it is clear that there is a need for an approach that will include attitudes about quality in the academic community by the students and employers, government institutions, and others.
The analyzed ranking methods represent one of the criteria according to which the authors of this paper selected and classified specific research indicators. The mentioned ranking methodologies represent a significant starting point for further research within this paper.
And the conclusions of the study? The conclusion must have:
Remember the objective of the study
This paper aims was to define and present a model for assessing the quality of HEIs in the technical-technological field and the decision-making system to support and adopt optimal management strategies for quality improvement.
There were two primary scientific objectives. The first objective was to develop a model for assessing the quality of HEIs in the technical-technological field from various stakeholders aspects. The first objective was to develop a system for decision support and make optimal decisions to improve individual quality indicators to define appropriate management strategies to improve quality. A model of KPIs was presented, i.e. a mathematical model was created that enabled the assessment of the quality of study programs and HEIs from different stakeholders' perspective and ranking and comparison of study programs and HEIs.
Main findings
Quality in the higher education sector is one of the current issues in the academic community. Monitoring and researching the higher education field, analyzing the literature and the current situation in the system of higher education in developing countries, it can be concluded that there is no single way to assess the quality of higher education institutions. This knowledge was a good starting point for the research presented in this paper. Accordingly, the findings include developing a system for quality assessment and higher education institutions ranking. Also, evaluating the relevance of key performance indicators (KPIs) of higher education institutions differs from different stakeholders' aspects. However, it is possible to develop a system for decision support and selection of the optimal strategy for improving study programs' performance and higher education institutions from the quality aspect.
Theoretical and practical implications
The practical implications include defining a decision support system that will enable the adoption of optimal decisions by the higher education institution's management team to improve the performance of study programs and the higher education institution. The presented system may enable benchmarking, and simulation and verification of different scenarios for improving the quality and performance of the higher education institutions.
Originality of the study
The originality of the research lies in the presented model that can be made available to government institutions and serve as a basis in the overall ranking and evaluation of higher education institutions with the possibility to develop a performance-based funding system. Also, other stakeholders can have an insight into the institution's performance for the sake of their needs and goals.
Study limitations
Within the research, certain limitations are primarily reflected in the selection and number of HEIs from the technical-technological field, which raises particular views and questions about whether the presented model applies to all HEIs in all academic areas. The sample size, the number of surveyed students and other stakeholders, the number of HEIs covered by the research, and the relevance of the data obtained from the management team of the HEIs being analyzed, also represent certain limitations.
Future lines of research
Directions for further research can be focused on covering all higher education institutions (faculties, academies and colleges) in developing counties, where the defined and proposed model can be implemented. Then, at the state level, the ranking of all higher education institutions can be carried out, based on the proposed model, and the National ranking system of higher education institutions can be proposed.
Further research can also be reflected through the concept of innovative education and smart university, which implies and is based on a new approach to education, learning and education strategy, distance learning, and the use of modern classrooms and laboratories.
Innovative education provides a framework for learning in new circumstances, based on trends and strategies of education and sustainable success.
Preparing students for a new role in the 21st century is the primary goal of intelligent education. This goal requires that the education system, from primary to higher education, be redesigned in line with the new demands and challenges of the 21st century.
The study's bibliography is weak. There is a lack of studies with
relevance to the topic and quality. Examples:
- Examining benchmark indicator systems for the evaluation of higher
education institutions
Authors (García-Aracil and Palomares-Montero, 2010) state that HEIs are undergoing essential changes involving the development of new roles and missions, with implications for their structure. They claim that there is difficulty establishing classification criteria for existing indicators, on which there is no consensus.
- Being highly internationalized strengthens your reputation: an
empirical investigation of top higher education institutions
In an educational context characterized by globalization, reputation constitutes a crucial issue for modern HEIs. The investigation results on the relationship between internationalization and reputation in top HEIs reveal that internationalization positively influences a university's reputation and moderates the relationship between its reputation and its institutional performance concerning research quality, teaching quality and graduate employability (Delgado-Márquez et al., 2013).
- Total quality management in UK higher education institutions
The authors (Kanji et al., 1999) examined how TQM principles and core concepts can be measured to provide a means of assessing the quality of different institutions on various aspects of their internal processes. It is found that the quality assurers could use the measurement method in the UK to assess the education quality of HEIs.
- E-learning success determinants: Brazilian empirical study
E-learning becomes a standard in the COVID 19 crisis, so it is essential to understand the impact of e-learning on society and its benefits. The authors (Cidral et al., 2018) aimed to find the determinants of user-perceived satisfaction, use, and personal impact of e-learning. The study proposed a theoretical model integrating theories of information systems' satisfaction and success in the e-learning systems. The model was empirically validated in HEIS through a quantitative structural equation modelling method. The drivers of user-perceived satisfaction are information quality, system quality, instructor attitude toward e-learning, diversity in assessment, and learner perceived interaction with others. System quality, use, and user-perceived satisfaction explain the personal impact.
- A model of factors affecting learning performance through the use of
social media in Malaysian higher education
The study (Al-Rahmi et al., 2018) attempted to mitigate the literature gap concerning social media use for active collaborative learning and engagement and its effect on Malaysia's research students' learning performance. The study concludes that overall, active collaborative learning and engagement through social media enriches students' learning activities and facilitates group discussions, and hence, their use should be encouraged in learning and teaching processes in HEIs.
Recently published studies must be included.
According to the Your comment, we have added recently published literature
- Adhikari, S.P. Transformational leadership practices in community school. Tribhuvan University Journal, 2019 33(1), 141-154.
- Agasisti, T.; Munda, G.; Hippe, R. Measuring the efficiency of European education systems by combining Data Envelopment Analysis and Multiple-Criteria Evaluation. Journal of Productivity Analysis 2019, 51(2), 105-124.
- Ahmed, O.A.A. The Effect Of Quality Of Higher Education System On The Compatibility Between The Skills Of Graduates And The Requirements Of The Labour Market In Egypt (Doctoral dissertation, Cardiff Metropolitan University), 2020.
- Alach, Z. Performance measurement and accountability in higher education: the puzzle of qualification completions. Tertiary Education and Management 2016, 22(1), 36 - 48.
- Al-Rahmi, W.M.; Alias, N.; Othman, M.S.; Marin, V.I.; Tur, G.A model of factors affecting learning performance through the use of social media in Malaysian higher education. Computers & Education 2018, 121, 59-72.
- Altbach, P.G. Rankings Season Is Here. Economic and Political Weekly 2010, 45(49), 14–17.
- Baporikar, N. Stakeholder approach for quality higher education. In Research Anthology on Preparing School Administrators to Lead Quality Education Programs (pp. 1664-1690), 2021. IGI Global.
- Becket, N.; Brookes, M. Evaluating quality management in university departments. Quality Assurance in Education 2016, 14(2), 123–142. http://doi.org/http://dx.doi.org/10.1108/QAE-11-2012-0046
- Biloshchytskyi, A.; Myronov, O.; Reznik, R.; Kuchansky, А.; Andrashko, Y.; Paliy, S.; Biloshchytska, S. A method to evaluate the scientific activity quality of HEIs based on a scientometric subjects presentation model. Восточно-Европейский журнал передовых технологий 2017, 6(2), 16-22.
- Blanco-Ramírez, G.; Berger, J.B. Rankings, accreditation, and the international quest for quality: Organizing an approach to value in higher education. Quality Assurance in Education 2014, 22(1), 88–104.
- Brookes, M.; Becket, N. Quality Management in Higher Education: A Review of International Issues and Practice. International Journal for Quality and Standards 2007, 1(1), 1–37. http://doi.org/10.1016/j.patrec.2008.02.004
- Case, J.M.; Marshall, D.; Linder, C. Being a student again: A narrative study of a teacher's experience. Teaching in Higher Education 2010, 15(4), 423-433.
- Chalmers, D. Defining Quality Indicators in the Context of Quality Models. Western Australia, 2008a.
- Choi, S. Identifying indicators of university autonomy according to stakeholders' interests. Tertiary education and management 2019, 25(1), 17-29.
- Chou, M.H.; Gornitzka, А. (Eds.). Building the knowledge economy in Europe: New constellations in European research and higher education governance. Edward Elgar Publishing. 2014. ISBN: 978 1 78254 528 6
- Cidral, W.A.; Oliveira, T.; Di Felice, M.; Aparicio, M. E-learning success determinants: Brazilian empirical study. Computers & Education 2018, 122, 273-290.
- Dai, L.; Li, J. Study on the quality of private university education based on analytic hierarchy process and fuzzy comprehensive evaluation method 1. Journal of Intelligent & Fuzzy Systems 2016, 31(4), 2241-2247.
- Daraio, C.; Bonaccorsi, A.; Simar, L. Rankings and university performance: A conditional multidimensional approach. European Journal of Operational Research 2015, 244(3), 918-930.
- Delgado-Márquez, B.L.; Escudero-Torres, M.A.; Hurtado-Torres, N.E. Being highly internationalized strengthens your reputation: an empirical investigation of top higher education institutions. Higher Education 2013, 66(5), 619-633.
- Dick, G.P.; Tarí. J.J. A Review of Quality Management Research in Higher Education Institutions. Kent Busines School Working Paper Series No 274 2013.
- Duerrenberger, N.; Warning, S. Corruption and education in developing countries: The role of public vs. private funding of higher education. International Journal of Educational Development 2018, 62, 217-225.
- Epifanić, V.; Urošević, S.; Dobrosavljević, A.; Kokeza, G.; Radivojević, N. Multi-criteria ranking of organizational factors affecting the learning quality outcomes in elementary education in Serbia. Journal of Business Economics and Management 2021, 22(1), 1-20. https://doi.org/10.3846/jbem.2020.13675
- Fernández-Cano, A.; Curiel-Marin, E.; Torralbo-Rodríguez, M.; Vallejo-Ruiz, M. Questioning the Shanghai Ranking methodology as a tool for the evaluation of universities: An integrative review. Scientometrics 2018, 116(3), 2069-2083.
- Findler, F.; Schönherr, N.; Lozano, R.; Stacherl, B. Assessing the impacts of higher education institutions on sustainable development—an analysis of tools and indicators. Sustainability 2019, 11(1), 59.
- Fleacă, E.; Fleacă, B.; Maiduc, S. Aligning strategy with sustainable development goals (SDGs): Process scoping diagram for entrepreneurial higher education institutions (HEIs). Sustainability 2018, 10(4), 1032.
- Franceško, M.; Nedeljković, J.; Živković, M.; Đurđić, S. Public and private higher education institutions in Serbia: Legal regulations, current status and opinion survey. European Journal of Education 2020, 55(4), 514-527.
- García-Aracil, A.; Palomares-Montero, D. Examining benchmark indicator systems for the evaluation of higher education institutions. Higher Education 2010, 60(2), 217-234.
- García-Peñalvo, F. J.; Corell, A.; Abella-García, V. Grande-de-Prado, M. Recommendations for Mandatory Online Assessment in Higher Education During the COVID-19 Pandemic. In Radical Solutions for Education in a Crisis Context (pp. 85-98). Springer, Singapore, 2021.
- Garde Sanchez, R.; Flórez-Parra, J.M.; López-Pérez, M.V.; López-Hernández, A.M. Corporate governance and disclosure of information on corporate social responsibility: An analysis of the top 200 universities in the shanghai ranking. Sustainability 2020, 12(4), 1549.
- Gontareva, I.; Borovyk, M.; Babenko, V.; Perevozova, I.; Mokhnenko, A. (2019). Identification of efficiency factors for control over information and communication provision of sustainable development in higher education institutions 2019.
- Gunn, A. Metrics and methodologies for measuring teaching quality in higher education: developing the Teaching Excellence Framework (TEF). Educational Review 2018, 70(2), 129-148.
- Kallio, K.M.; Kallio, T.J.; Grossi, G. Performance measurement in universities: ambiguities in the use of quality versus quantity in performance indicators. Public Money & Management 2017, 37(4), 293-300.
- Kanji, G.K.; Malek, A.; Tambi, B.A. Total quality management in UK higher education institutions. Total Quality Management 1999, 10(1), 129-153.
- Kayani, M. Analysis of Socio-Economic Benefits of Education in Developing Countries: A Example of Pakistan. Bulletin of Education and Research 2017, 39(3), 75-92.
- Khashab, B.; Gulliver, S.R.; Ayoubi, R.M. A framework for customer relationship management strategy orientation support in higher education institutions. Journal of Strategic Marketing 2020, 28(3), 246-265.
- Kooli, C. Governing and managing higher education institutions: The quality audit contributions. Evaluation and program planning 2019, 77, 101713.
- Krüger, K.; Parellada, M.; Samoilovich, D.; Sursock, A. Governance reforms in European university systems. The Case of Austria, Denmark, Finland, France, the Netherlands and Portugal. Cham: Springer, 2018.
- Kryscynski, D.; Coff, R.; Campbell, B. Charting a path between firm‐specific incentives and human capital‐based competitive advantage. Strategic Management Journal 2021, 42(2), 386-412.
- Leal Filho, W.; Shiel, C.; Paço, A.; Mifsud, M.; Ávila, L.V.; Brandli, L.L.; ... Caeiro, S. Sustainable Development Goals and sustainability teaching at universities: Falling behind or getting ahead of the pack?. Journal of Cleaner Production 2019, 232, 285-294.
- Martin, M. Using indicators in higher education policy: between accountability, monitoring and management. In Research Handbook on Quality, Performance and Accountability in Higher Education. Edward Elgar Publishing, 2018.
- Martin, M.; Sauvageot, C. Constructing an indicator system or scorecard for higher education: A practical guide. Paris, France: UNESCO - International Institute for Educational Planning. ISBN: 978-92-803-1329-1, 2011.
- Massy, W.F.; Sullivan, T.A.; Mackie, C. 2013. Improving measurement of productivity in higher education. Change: The Magazine of Higher Learning 2013, 45(1): 15–23.
- Mensah, J. Improving Quality Management in Higher Education Institutions in Developing Countries through Strategic Planning. Asian Journal of Contemporary Education 2020, 4(1), 9-25.
- Miller, B. A. Assessing organizational performance in higher education. John Wiley & Sons. ISBN 978-0-7879-8640-7. 2016.
- Nazari-Shirkouhi, S.; Mousakhani, S.; Tavakoli, M.; Dalvand, M.R.; Šaparauskas, J.; Antuchevičienė, J. Importance-performance analysis based balanced scorecard for performance evaluation in higher education institutions: an integrated fuzzy approach. Journal of Business Economics and Management 2020, 21(3), 647-678.
- Nurcahyo, R.; Wardhani, R.K.; Habiburrahman, M.; Kristiningrum, E.; Herbanu, E.A. Strategic formulation of a higher education institution using balance scorecard. In 2018 4th International Conference on Science and Technology (ICST) (pp. 1-6). IEEE, 2018.
- Owens, T. L. Higher education in the sustainable development goals framework. European Journal of Education 2017, 52(4), 414-420.
- Ramasamy, N.; Rajesh, R.; Pugazhendhi, S.; Ganesh, K. Development of a hybrid BSC-AHP model for institutions in higher education. International journal of enterprise network management 2016, 7(1), 13-26.
- Rosa, M.J.; Sarrico, C.S.; Amaral. A. Implementing Quality Management Systems in Higher Education Institutions. In Quality Assurance and Management, edited byMehmet Savsa, 129–46. Rijeka: INTECH, 2012.
- Roth, M.G.; McAndrew, W.P. To each according to their ability? Academic ranking and salary inequality across public colleges and universities. Applied Economics Letters 2018, 25(1), 34-37.
- Rymarzak, M.; Marmot, A. Higher education estate data accountability: the contrasting experience of UK and Poland. Higher Education Policy 2020, 33(1), 179-194.
- Sam, V. Overeducation among graduates in developing countries: What impact on economic growth?. 2018.
- Santos, G.; Marques, C.S.; Justino, E.; Mendes, L. Understanding social responsibility's influence on service quality and student satisfaction in higher education. Journal of cleaner production 2020, 256, 120597.
- Soares, M.C.; Novaski, O.; Anholon, R. SERVQUAL model applied to higher education public administrative services. Brazilian Journal of Operations & Production Management 2017, 14(3), 338-349.
- Soewarno, N.; Tjahjadi, B. Mediating effect of strategy on competitive pressure, stakeholder pressure and strategic performance management (SPM): evidence from HEIs in Indonesia. Benchmarking: An International Journal
- Vernon, M.M.; Balas, E.A.; Momani, S. Are university rankings useful to improve research? A systematic review. PloS one 2018, 13(3), e0193762.
- Wang, D.D. Performance-based resource allocation for higher education institutions in China. Socio-Economic Planning Sciences 2019, 65, 66-75.
- Woodall, T.; Hiller, A.; Resnick, S. Making sense of higher education: students as consumers and the value of the university experience. Studies in Higher Education 2014, 39(1), 48-67.
- Yusof, N.; Hashim, R.A.; Valdez, N.P.; Yaacob, A. Managing diversity in higher education: A strategic communication approach. Journal of Asian Pacific Communication 2018, 28(1), 41-60.
- Zwain, A.A.A.; Lim, K.T.; Othman, S.N. TQM and academic performance in Iraqi HEIs: associations and mediating effect of KM. The TQM Journal 2017.

Reviewer 2 Report
The title is long and not very appealing.
The Abstract must have the following logic: Purpose; Design / methodology / approach; Findings; Practical implications; Originality / value
Its introduction is very weak. Lacks bibliography. There are only two authors cited!
The literature GAP needs to be clarified and reinforced. The purpose of the study is also unclear. I recommend redoing the entire introduction:
1- Make a frame for the reader
2- Issues of the topic under analysis
3- Evidence of the GAP of the literature based on the literature
4- Purpose of the study
5- Originality of the study
6- What are the expected results (to captivate the reader)
7- The last paragraph should briefly describe what the reader can read in the following sections.
After the introduction, there has to be a literature review.
Does the study have no methodology? After the literature review, the methodology has to come. The methodology has to be detailed. Is the study quantitative or qualitative? What method is used? Is it a bibliometric review? This needs to be justified and explained.
In discussing the results, the authors do not confront them with the literature.
And the conclusions of the study? The conclusion must have:
- Remember the objective of the study
- Main findings
- Theoretical and practical implications
- Originality of the study
- Study limitations
- Future lines of research
The study's bibliography is weak. There is a lack of studies with relevance to the topic and quality. Examples:
- Examining benchmark indicator systems for the evaluation of higher education institutions
- Being highly internationalized strengthens your reputation: an empirical investigation of top higher education institutions
- Total quality management in UK higher education institutions
- E-learning success determinants: Brazilian empirical study
- A model of factors affecting learning performance through the use of social media in Malaysian higher education
Recently published studies must be included.
Good luck to the authors for the publication of the study!
Author Response

(The authors gave the same response as above.)

Round 2
Reviewer 2 Report
- This title “3. Definition of a set of indicators” must-have methodology.
- The methodology remains to be detailed. How was the questionnaire applied? Has it been applied before? If so, they must indicate the source.
- How was the questionnaire validated? What is the response rate?
- Why did you use this methodology and not another? What is the importance of using this method in this study? This has to be explained very well.
- If it is a quantitative study, the hypotheses must be formulated in the literature review.
- They must make the research model at the end of the literature review.
- In this title “2. Materials and Methods ”I recommend including“ Literature review ”.
- Images 6 and 7 must be worked on. Do they look like screen prints?
- The discussion of results must be separated from the conclusion.
- I recommend that you structure the conclusion as follows:
7. Conclusions
7.1. Main findings
7.2.Theoretical implications
7.3. Practical implications
7.4. Study limitations
7.5. Future lines of research
Good luck with the publication!
Author Response
This title "3. Definition of a set of indicators" must-have methodology.
The title of the section has been renamed according to Your suggestion, in the following manner:
- Research methodology and a set of indicators definition
With separated subsections:
4.1 Research methodology
4.2 Set of indicators definition
The methodology remains to be detailed. How was the questionnaire applied? Has it been applied before? If so, they must indicate the source.
Based on Your suggestion, we have tried to give a detailed methodology explanation and present it in the 4.1 Research methodology section.
The developed questionnaire was sent to the representatives' e-mail addresses of all accredited HEIs in the Republic of Serbia from the techno-technological (TT) field, and it has not been applied before.
How was the questionnaire validated? What is the response rate?
Based on Your suggestion, we have tried to give a detailed methodology explanation and present it in the 4.1 Research methodology section.
A questionnaire validity has been established using a panel of experts employed in the Republic of Serbian Ministry of Education, Science and Technological Development, who possess significant HEIs management experience.
The distribution of the number of accredited HEIs according to the type of institutions from the TT field to whom the questionnaire was forwarded is as follows: 10 universities, 46 faculties and 26 higher technical schools. The distribution of responses received is as follows: 8 questionnaires from universities, 38 questionnaires from faculties and 19 questionnaires from higher technical schools. Consequently, the response rate is about 79% in total.
Why did you use this methodology and not another? What is the importance of using this method in this study? This has to be explained very well.
Based on Your suggestion, we have tried to give a detailed methodology explanation and present it in the 4.1 Research methodology section.
The methodology was developed for the Republic of Serbian Ministry of Education, Science and Technological Development needs and implemented as a pilot project, and this is why the higher number of HEIs responded. The importance of methodology reflects in the fact that there is a need for a methodology that will better take into account the needs of different stakeholders, be more balanced and better suited to regional, medium and small universities since the existing methodologies favour the largest HEIs. In this way, the limitations of existing methodologies for evaluating and ranking universities (discussed in the introductory part)could be overreached.
If it is a quantitative study, the hypotheses must be formulated in the literature review.
Based on Your suggestion, we have introduced two hypotheses, which are incorporated into the research model.
Based on the literature review and presented methodology following hypothesis have been formulated:
H1: It is possible to develop a system for quality assessment and ranking of higher education institutions.
H2: It is possible to develop a system to support decision-making and selection of the optimal strategy for improving the performance of study programs and higher education institutions from the aspect of quality.
They must make the research model at the end of the literature review.
According to Your suggestion, we have made the research model proposal.
Figure 1. Research model
According to Your suggestion, we have made the research model proposal.
In this title "2. Materials and Methods" I recommend including "Literature review".
According to Your suggestion, we have separated this section into sections and named them 2. Literature review, and 3. Materials and Methods, which is highlighted with red colour in the revised version.
Images 6 and 7 must be worked on. Do they look like screen prints?
Accordng to Your suggestions, the quality of figures 6 and z has been improved.
Figure 6.The number of Q1 and Q2 publications of the University of Kragujevac.
Figure 7.The number of Q1 and Q2 publications in Engineering. University of Kragujevac
The discussion of results must be separated from the conclusion.
We have separated the discussion and conclusion according to Your comments.
I recommend that you structure the conclusion as follows:
The conclusion has been structured according to Your suggestion.
- Conclusions
8.1. Main findings
Definition of the model for performance-based evaluation in HE (institutions and study programs) has considerable importance for different target groups:
(1) The academic community and the universities in a specific country. Develop-ing a system for measuring performance and multi-dimensional ranking of study programs and institutions should contribute to a better, higher quality, more efficient, market-oriented and socially responsible management of study programs and universities. Ranking of study programs and institutions will, through KPIs, enable focus on critical processes, process benchmarking, comparison and thus improving key processes at universities in Serbia which should contribute to higher levels of education, research, development and innovation processes, higher terms of the internationalisation as well as co-operation with industry locally and in the region. On the other hand, all this affects the definition and redefine institutional strategies.
(2) Students. According to the ranking of study programs and institutions, students will have opportunities to make choices that suit them best, bearing in mind the set of performance, which would indicate essential parameters (for example, the number of unemployed graduates is a study program).
(3) Industry and business. Business entities would have an overview of different orientations and parameters defined study programs, market orientation and quality. In this way, it is possible to achieve feedback between HEIs and industry.
(4) National Employment Service and the Republic Institute for Statistics of focus group would benefit from the access to real data, their organised monitoring and better connections with HEIs and businesses (today, for example), it is not possible to generate information on the number of unemployed graduates originated from individual institutions,
(5) Government and state institutions (policymakers) can use the defined set of performance to create the legal framework and recommendations for funding or financial models that should be incorporated indicators measuring quality educational process.
This paper aims to define and present a model for assessing the quality of HEIs in the technical-technological field and the decision-making system to support and adopt optimal management strategies for quality improvement.
There were two primary scientific objectives. The first objective was to develop a model for assessing the quality of HEIs in the technical-technological field from various stakeholders' aspects. The first objective was to develop a system for decision support and make optimal decisions to improve individual quality indicators to define appropriate management strategies to improve quality. A model of KPIs was presented, i.e. a mathematical model was created that enabled the assessment of the quality of study programs and HEIs from different stakeholders' perspective and ranking and comparison of study programs and HEIs.
Quality in the HE sector is one of the current issues in the academic community. Monitoring and researching the HE field, analyzing the literature and the current situation in the system of HE in developing countries, it can be concluded that there is no single way to assess the quality of HEIs. This knowledge was a good starting point for the research presented in this paper. Accordingly, the findings include developing a system for quality assessment and HEIs ranking. Also, evaluating the relevance of KPIs of HEIs differs from different stakeholders' aspects. However, it is possible to develop a system for decision support and selection of the optimal strategy for improving study programs' performance and HEIs from the quality aspect.
The realisation of more comprehensive and specific objectives needs to provide public value management, moving from bureaucracy to a market approach while providing HE to serve society better
The presented model has several advantages: in the first place, this model introduces different perspectives (these perspectives are different that perspectives in existing models). In the second place, this model introduces weights for all KPIs. Finally, this model enables ranking and benchmarking between institutions. This model considers all stakeholders' views and the number of indicators from a specific environment, so it is more friendly and accurate for smaller and regional Universities. Finally, we introduced a software solution that enables HEIs to monitor, track, and improve their indicators and essential processes. The suggested model has great potential because as many universities use this model, benchmarking will be broader and more functional.
8.2.Theoretical implications
The theoretical implications reflect in the defined model for quality assessment and performance assessment (based on KPIs) of study programs and HEIs while considering a complex group of stakeholders requirements. The model was following novelty: definition of specific perspectives, unique set of KPIs (respecting demands of all stakeholders), Introduction of weights for specifis KPIs. The presented matehematical model with 9 steps using GA optimization is important contribution of this mansucript. This model opens the way to define the methodology for measuring, monitoring and improving the HEIs and the education system quality.
8.3. Practical implications
The practical implications include defining a decision support system that will enable the adoption of optimal decisions by the HEIs management team to improve the performance of study programs and the HEIs. The presented system may enable benchmarking, and simulation and verification of different scenarios for improving the quality and performance of the HEIs.
The originality of the research lies in the presented model that can be made available to government institutions and serve as a basis in the overall ranking and evaluation of HEIs with the possibility to develop a performance-based funding system. Also, other stakeholders can have an insight into the institution's performance for the sake of their needs and goals.
8.4. Study limitations
Within the research, certain limitations are primarily reflected in the selection and number of HEIs from the technical-technological field, which raises particular views and questions about whether the presented model applies to all HEIs in all academic areas. The sample size, the number of surveyed students and other stakeholders, the number of HEIs covered by the research, and the relevance of the data obtained from the management team of the HEIs being analyzed, also represent certain limitations.
8.5. Future lines of research
Directions for further research can be focused on covering all HEIs (faculties, academies and colleges) in developing counties, where the defined and proposed model can be implemented. Then, at the state level, the ranking of all HEIs can be carried out, based on the proposed model, and the National ranking system of HEIs can be proposed.
Further research can also be reflected through the concept of innovative education and smart university, which implies and is based on a new approach to education, learning and education strategy, distance learning, and the use of modern classrooms and laboratories.
Innovative education provides a framework for learning in new circumstances, based on trends and strategies of education and sustainable success.
Preparing students for a new role in the 21st century is the primary goal of intelligent education. This goal requires that the education system, from primary to HE, be redesigned in line with the new demands and challenges of the 21st century.

Round 3
Reviewer 2 Report
The authors made an effort to improve the article.
I think the article can be accepted after:
- The discussion of the results has to be developed
- Theoretical implications have to be developed
- Practical implications have to be developed
- Study limitations have to be developed
Author Response
Dear reviewer, according to Your comments, we have made changes marked with red colour.
I think the article can be accepted after:
The discussion of the results has to be developed
Dear reviewer, according to Your comment, we have made changes:
- Discussion
The analysed ranking methods represent one of the criteria according to which the authors of this paper selected and classified specific research indicators. The paper [84] contributes to creating a ranking list generation that improves the art's current state on methodological and substantive bases with two contributions: integrating new information and using new ranking techniques. The main criticisms are mono dimensionality, statistical robustness, dependence on university size and subject mix and lack of consideration. The authors [85] conducted a systematic review of university ranking systems to evaluate ranking systems' usefulness and identify opportunities to support and improve research quality and performance. They concluded that future research should evaluate three research outcomes: scientific impact, economic outcomes and public health impact. The author [86] presented a developed performance-based method for a central planner to allocate research funding to different universities to stimulate the research output. Although this paper focuses on China's HE system, the research framework is general and can be applied to other countries. Each of the mentioned systems looks at the HEIs individual quality segments differently and evaluates them differently. These facts indicate that the issues reflect the vaguely defined criteria according to which the indicators should be selected. According to all the above, it is clear that there is a need for an approach that will include attitudes about quality in the academic community by the students and employers, government institutions, and others. The mentioned ranking methodologies represent a significant starting point for further research within this paper.
The presented model for quality assessment and performance assessment (based on KPIs) of study programs could be used to measure, monitor and to improve the HEIs and the education system quality on the one hand and essential tool for benchmarking, decision support and finally, a step towards financing educational systems according to the performances. The results for all the considered KPI groups divided by dimensions (Institution, Teaching, Science, Service Users, Employers/Economic, Country/Society) were obtained by applying the presented solution. Figure 2a shows the ranks of KPIs within the Institution group. The Institution group contains nine KPIs, where the most influential is indicator KPIID1 (An average grade from previous educational level), then indicator KPIID3 (% of maximal 60 ECTS from the previous year), and indicator KPIID2 (Total number of students in the 1st year). These three indicators are ranked the highest compared to the remaining six indicators in this group. Indicators KPIID9 (Income from students), indicator KPIID4 (Number of foreign students), and indicator KPIID7 (Financing of science) have the lowest rank, which is the least important from a quality aspect, but undoubtedly crucial from the HEIs business aspect. Indicators KPIID5 (Percentage of graduates), KPIID6 (Finances of HEI [total income]), and KPIID8 (Income from students' fees) are of medium importance. These results show that the previous educational level quality plays an essential role in HIEs current quality performance.
Based on the presentation in Figure 2b, in the group Teaching, it can be concluded that the KPITD1 (Number of study programs) has the highest value or rank, while KPITD3 (Evaluation of study program [students evaluation]) has the lowest rank. Other indicators, KPITD2 (Number of students in lecturing groups), KPITD4 (Evaluation of teaching process program [students evaluation]) and KPITD5 (Student internship), have average values that are half as significant as the highest-ranked KPITD1. These results show that the study programs diversity presents a significant factor in HEIs quality determination.
Figure 3a presents a range of KPIs within the Science group, which contains six indicators with their values. The values of KPISD4 (Academic staff mobility) represent the highest-ranked indicator, while indicator KPISD5 (Students' mobility) has a slightly lower value and rank. The remaining four indicators, KPISD1 (Published manuscripts), KPISD2 (Number of publication at SCI, SSCI), KPISD3 (Number of books), and KPISD6 (Publications in international cooperation), have equal values and the lowest rank compared to higher-ranked KPIs. Although they have a lower rank and not so much impact on HEs' quality, they are significant since they can be applied to assess teaching staff quality. These results show that the academic staff and students mobilities are essential since these mobilities lead to the exchange of knowledge and culture at the international level.
The Stakeholders group (Figure 3b) contains four indicators, of which the KPISSD1 (The average duration of studies) has the highest ranking in the group, while the lowest-ranked indicator is KPISSD2 (Learning outcomes in graduate students). Indicator KPISSD3 (An unemployment rate of graduates) and indicator KPISSD4 (Students scholarship [from industry]) are in the middle compared to other KPIs in the group. These results show that the average duration of studies is essential since the labour market needs to be provided with a trained workforce in the short term.
The rank of KPIs within the group Employers/Business is presented in Figure 4a. The highest rank has KPIED1 (Projects with business entities). The KPIED2 (Number of BSc and MSc realised with business) is ranked lower, while the KPIED3 (Scientific manuscripts with business) is ranked the lowest. These results show that the most critical performance is related to the cooperation with business entities since such cooperation produces new research ideas, which can contribute to sustainable HEIs and business entities development.
The ranking of KPIs within the State/Society group is presented in Figure 4b. KPISS3 (Public lectures) is ranked highest, KPISS1 (Participation in national projects) is of medium importance, while KPISS2 (Number of projects financed by the state) has the lowest rank. These results show that public lectures have a significant impact since they lead to improved HEIs public image.
Based on the Pareto front values (Figure 5), it can be determined which HEI is first in the rank. The values of the best ranked HEIs' KPIs were compared with the values determined in Step 2 of the proposed Algorithm. Figure 6 presents a comparison of KPIs' values between the best ranked HEIs, randomly selected HEI values of KPIs, and average values of KPIs at the level of all considered HEIs. By optimising specific KPIs, the performance of the HE process can be improved. Improving KPIs will ensure a better ranking of HEIs and improve performance in terms of quality. This model's main advantage has several advantages comparing the existing evaluation model: it has more dimensions, a novel mathematical approach and more suitable for general HEIs.
Theoretical implications have to be developed
Dear reviewer, according to Your comment, we have made changes:
8.2. Theoretical implications
The theoretical implications reflect in the defined model for quality assessment and performance assessment (based on KPIs) of study programs and HEIs while considering a complex group of stakeholders requirements. The model was the following novelty: definition of specific perspectives, a unique set of KPIs (respecting demands of all stakeholders), Introduction of weights for specific KPIs. The presented mathematical model with nine steps using GA optimisation is an essential contribution of this manuscript. This model opens the way to define the methodology for measuring, monitoring and improving the HEIs and the education system quality.
As there was no clearly defined methodological approach that combined the application of business intelligence tools for HEIs process and performance management, the paper has theoretical implications which are reflected in:
(1) defining models to support decision-making on quality objectives and business performance in HEIs,
(2) determining the measure of execution of processes, subprocesses and their KPIs based on the realised results in the form of performance for small and medium HEIs,
(3) determining and optimising specific KPIs based on the observed performance that needs to be improved and,
(4) predicting HEIs performance improvements based on established optimal KPI improvements.
Practical implications have to be developed
Dear reviewer, according to Your comment, we have made changes:
8.3. Practical implications
The practical implications include defining a decision support system that will enable the adoption of optimal decisions by the HEIs management team to improve the performance of study programs and the HEIs. The model enables identifying the currently most influential KPIs HEIs, so managers and decision-makers are allowed to select KPIs for optimal improvement. By determining the optimal improvements, it is possible to identify and solve business problems by applying an appropriate business strategy and thus improve overall HEIs performance.
The presented system may enable benchmarking. It enables each HEI to compare their processes or subprocesses with the rank of processes or subprocesses of all other HEIs based on the achieved performance and the quality of the analysed service processes. This enables for HEIs to quantify the measure of achieving the set business goals.
Also, the solution enables simulation and verification of different scenarios for improving the system's quality and performance, which can have highly positive effects on the management and improvement of higher education.
Consequently, it may be concluded that the integrated ranking solution practical implications reflect in providing the opportunities for organisations to report, identify bottlenecks and influencing factors, solve dynamic HEIs problems, monitor HEIs processes, learn from leading HEIs, compare processes and subprocesses in different HEIs.
The originality of the research lies in the presented model that can be made available to government institutions and serve as a basis in the overall ranking and evaluation of HEIs with the possibility to develop a performance-based funding system. Also, other stakeholders can have an insight into the institution's performance for the sake of their needs and goals.
Study limitations have to be developed
Dear reviewer, according to Your comment, we have made changes:
8.4. Study limitations
Within the research, certain limitations are primarily reflected in the selection and number of HEIs from the technical-technological field, which raises particular views and questions about whether the presented model applies to all HEIs in all academic areas. Furthermore, the limitations are primarily related to selecting specific HEIs (small and medium-sized), so the question arises whether the model applies to large HEIs. The model was tested on data obtained from organisations in the Republic of Serbia so that the results are not valid for decision-making in HEIs outside its borders.
In addition to the above limitations, there are also limitations related to the validity of data obtained from HEIs. The data were obtained from HEIs managers and decision-makers, so there is a risk that the obtained data are not objective in reviewing the estimated values of processes, subprocesses, KPIs and HEIs performance.
